# A Multilevel Low-Rank Newton Method with Super-linear Convergence Rate and its Application to Non-convex Problems

**Nick Tsipinakis**  *nikolaos.tsipinakis@unidistance.ch*
*Department of Mathematics and Computer Science*
*UniDistance Suisse*
*Brig, Switzerland*

**Panagiotis Tigas**  *ptigas@robots.ox.ac.uk*
*Department of Computer Science*
*Oxford University*
*Oxford, UK*

**Panos Parpas**  *panos.parpas@imperial.ac.uk*
*Department of Computing*
*Imperial College London*
*London, UK*

**Reviewed on OpenReview:** *https://openreview.net/forum?id=PKakPzVVja*

## Abstract

Second-order methods can address the shortcomings of first-order methods for the optimization of large-scale machine learning models. However, second-order methods have significantly higher computational costs associated with the computation of second-order information. Subspace methods that are based on randomization have addressed some of these computational costs as they compute search directions in lower dimensions. Even though super-linear convergence rates have been empirically observed, it has not been possible to rigorously show that these variants of second-order methods can indeed achieve such fast rates. Also, it is not clear whether subspace methods are efficient for non-convex settings. To address these shortcomings, we develop a link between multigrid optimization methods and low-rank Newton methods that enables us to prove the super-linear rates of stochastic low-rank Newton methods rigorously. Our method does not require any computations in the original model dimension. We further propose a truncated version of the method that is capable of solving high-dimensional non-convex problems. Preliminary numerical experiments show that our method has a better escape rate from saddle points compared to the state-of-the-art first-order methods and thus returns lower training errors.

## 1 Introduction

When it comes to applying second-order optimization methods in machine learning, there are two open questions: 1. Can second-order methods be implemented efficiently? 2. Can second-order methods outperform standard first-order methods for traditional ML metrics such as generalization error? Regarding the latter question, several recent articles have argued the efficacy of second-order methods in deep learning (Singh & Alistarh, 2020; Pascanu & Bengio, 2013; Dauphin et al., 2014), reinforcement learning (Wu et al., 2017) and variational inference (Regier et al., 2017) (to name but a few). Given the recent promising experimental results obtained via second-order methods, the first question regarding the efficiency of these methods has become even more urgent. In this paper we provide some answers to the first question.

The development of randomized methods such as sketching and sub-sampling has enabled the application of second-order methods to large scale ML models. Super-linear or linear-quadratic (composite) convergence for sub-sampled methods has been shown in various recent works (Berahas et al., 2017; Bollapragada et al., 2019; Erdogdu & Montanari, 2015; Pilanci & Wainwright, 2017; Roosta-Khorasani & Mahoney, 2019; Na et al., 2023). But all the existing methods require $O(n^3)$ operations to compute the Newton direction. Stochastic multilevel or sub-space methods, such as the one proposed in this paper, have $O(nN)$ cost for computing and storing the Hessian, and $O(n^2N)$ cost for computing the Newton direction (where $N \ll n$ is the dimension of the sub-space). However, it is unclear if stochastic multilevel methods can still converge super-linearly when the hessian is approximated via a random method. Although it is clear from numerical experiments that randomized Newton methods can obtain super-linear convergence, rigorous proof under general conditions has not appeared before Hanzely et al. (2020); Gower et al. (2019).

In terms of the theoretical contributions of this paper, the algorithms of Pilanci & Wainwright (2017) and Tsipinakis & Parpas (2024) are particularly relevant since they are analyzed using self-concordant functions. In Pilanci & Wainwright (2017), restrictive assumptions are made regarding the Hessian. In particular, the authors assume that the square root of the Hessian is available or easily computable, and the method requires access to the full Hessian and both the proof and hence the algorithm cannot be extended to the non-convex case. In this paper we relax all these assumptions without compromising the convergence rate. In Tsipinakis & Parpas (2024), super-linear convergence was established, but the algorithm cannot be applied to the non-convex case without non-trivial modifications. This paper links multilevel optimization methods and randomized Newton methods. We exploit this link and establish the global convergence of the algorithm and a local super-linear convergence rate for self-concordant functions. Because of the way the low-rank Hessian approximation is performed, our method can easily be extended to the non-convex case and to very high dimensions (even when the Hessian is dense).

Regarding the stochastic subspace methods, the work of Gower et al. (2019) is quite relevant, however the method only achieves a linear rate, whereas we achieve a super-linear rate for the same iteration complexity. In addition, the method in Gower et al. (2019) cannot be extended to the non-convex case without non-trivial modifications. In Hanzely et al. (2020), the authors develop a related method that is based on the cubic Newton method. But the method has a linear rate, and it is unclear if the method can be efficiently applied to non-convex problems. Shang et al. (2024) develop an accelerated double-sketching subspace Newton method that also attains a local linear rate. Furthermore, in Fuji et al. (2025) and Cartis et al. (2025), the authors establish global convergence for subspace-regularized Newton methods. However, fast local convergence rates are not available, and these methods have only been tested on relatively small non-convex problems.

Here, in order to apply the Newton method to non-convex problems, we take a truncated SVD of the Hessian to keep the $N + 1$ most informative eigenvalues and replace the rest with the $(N + 1)^{\text{th}}$ eigenvalue while negative eigenvalues are replaced by their absolute value. Therefore, as in the convex case, enforcing the Hessian matrix to be positive definite and premultiplying the negative gradient by its inverse, we perform a local change of coordinates and thus we should expect an accelerated convergence behavior compared to the first-order methods. In addition, when dealing with non-convex functions, the fact that we do not allow for sufficiently small eigenvalues means that slow and flat manifolds around saddle points will turn into saddles whose eigenvalues are large and we therefore achieve a faster escape rate from the unstable region of the saddle points. Similar approaches when computing the Newton direction have already been explored (O'Leary-Roseberry et al., 2019; Paternain et al., 2019; Erdogdu & Montanari, 2015). However, they all require the computation of the full Hessian matrix and hence they are inefficient for high dimensional problems. A work that scales to high dimensional problems which also approximates the Hessian matrix was explored by Marteau-Ferey et al. (2019). The difference to our approach is that it regularizes the Hessian matrix instead of performing a Truncated SVD. However, selecting the regularization parameter such that the Hessian matrix becomes positive definite is a difficult task and thus the method is suitable to convex problems. In addition, their method has limited applicability since it relies on the existence of an $\ell_2$ regularizer to work. Such an assumption is not present in this paper.

To this end, all the aforementioned advantages of our approach are demonstrated in our numerical experiments. We show that our method is efficient in terms of computational requirements and it can be applied to models with millions of parameters even when the Hessian is dense. In particular, the experiments suggest

that the proposed methods are well-suited for problems with low-rank Hessian matrix—an assumption that is satisfied in most practical problems, especially in non-convex settings. Our numerical experiments suggest that the proposed method behaves similarly to the cubic Newton method (in terms of efficiency and ability to escape saddle points) but with significantly less computational requirements. Moreover, we apply the method to the minimization of a deep autoencoder model, which is known to exhibit numerous flat regions and saddle points. The results demonstrate that the proposed method escapes saddle points and flat regions significantly more efficiently than Adam, despite constructing its iterates in a much smaller subspace.

## 2 Background and Method

In this section, we briefly discuss the main components of multilevel optimization methods, and introduce the coarse-grained model, which we later modify to compute search directions. In the optimization literature, multilevel methods are also referred to as multigrid methods. Since there is no notion of "grid" in the proposed algorithm we use the term multilevel, but in the context of this paper the two terms are equivalent. Multilevel optimization methods have two components: (1) A hierarchy of coarse models, and (2) Linear prolongation and restriction operators used to "transfer" information up and down the hierarchy of models. In this section, we describe these two components; however, for simplicity, we consider a two-level hierarchy. Our results can be extended to more than two levels without substantial changes.

### 2.1 The Coarse-grained Model and Main Assumptions

We first develop the proposed methodology for a convex optimization model. The non-convex case is discussed in Section 3.1.1. Consider the following model,

$$\mathbf{x}^* = \arg\min_{\mathbf{x} \in \mathbb{R}^n} f(\mathbf{x}). \tag{1}$$

where we assume that the function $f$ is a strictly convex and self-concordant. Moreover, it has a closed sub-level set and is bounded below so that $\mathbf{x}^\star$ exists. The definition and properties of self-concordant functions can be found in appendix A, and we refer the interested reader to Boyd & Vandenberghe (2004); Nesterov (2004); Nesterov et al. (2018); Nesterov & Nemirovskii (1994) for additional background information. We will adopt the terminology used in the multilevel literature and call the model in equation 1 the *fine model*. We will also assume that we can construct a lower dimensional model called the *coarse model*. In traditional multilevel methods the coarse model is typically derived by varying a discretization parameter. In machine learning applications a coarse model can be derived by varying the number of pixels in image processing applications (Galun et al., 2015), or by varying the dictionary size and fidelity in statistical pattern recognition (see e.g. Hovhannisyan et al. (2016) for examples in face recognition, and background extraction from video). We denote the coarse model as,

$$\mathbf{y}^* = \arg\min_{\mathbf{y} \in \mathbb{R}^N} F(\mathbf{y}). \tag{2}$$

We assume that $N < n$ and typically $N \ll n$ and that $F$ is also a strictly convex, self-concordant function. The property that $N < n$ justifies the use of the terms *coarse-grained* or reduced order model. The role of the coarse model is to help generate high quality search directions from the current incumbent point $\mathbf{x}_k$ but at a reduced computational cost. Let $\mathbf{R}_k \in \mathbb{R}^{N \times n}$ be the *restriction operator* with which one may transfer information from the fine to coarse model and thus we define the initial point in the coarse model as $\mathbf{y_0} := \mathbf{R}_k \mathbf{x}_k$ at iteration $k$. For simplification purposes from now onwards we will omit the subscript $k$ from the restriction operator. In order for the search direction to be useful (e.g. a descent direction), we assume that *first-order coherency condition* below holds,

$$\mathbf{R}\nabla f(\mathbf{x}_k) = \nabla F(\mathbf{y_0}), \tag{3}$$

See Wen & Goldfarb (2009) for a discussion of the first-order coherency condition in multilevel optimization. In addition to the restriction operator we also assume that a prolongation operator, $\mathbf{P} \in \mathbb{R}^{n \times N}$ is available. We make the following assumption about these two operators.

**Assumption 1.** *The restriction and prolongation operators* $\mathbf{R}$ *and* $\mathbf{P}$ *satisfy,* $\mathbf{P} = \sigma\mathbf{R}^T$, *where* $\sigma > 0$, *and* $\mathbf{P}$ *has full column rank, i.e.,* $\mathrm{rank}(\mathbf{P}) = N$.

The assumptions regarding the restriction and prolongation operators are standard (Briggs et al., 2000) and, without loss of generality, we assume that $\sigma = 1$. A simple way to construct the prolongation and restriction operators, which satisfies Assumption 1, arises from the naive Nyström method (see Drineas & Mahoney (2005) for an introduction). In particular, we construct $\mathbf{R}$ and $\mathbf{P}$ as described in the definition below.

**Definition 1.** *Let* $S_n = \{1, 2, \ldots, n\}$ *and denote* $S_N \subset S_n$, *with the property that the elements* $N < n$ *are uniformly selected by the set* $S_n$ *without replacement. Furthermore, assume that* $s_i$ *is the* $i^{th}$ *element of* $S_N$. *Then the prolongation operator* $\mathbf{P}$ *is generated as follows: the* $i^{th}$ *column of* $\mathbf{P}$ *is the* $s_i$ *column of* $\mathbf{I}_{n \times n}$ *and, furthermore, we set* $\mathbf{R}$ *as* $\mathbf{P}^T$.

Beyond definition 1, several ways to select the restriction operators are available in the literature, e.g., Gaussian, 1- or s-hashing, and sampling matrices; see Cartis et al. (2022). However, for our experiments, we prefer uniform sampling for constructing these operators because it yields efficient iterations by avoiding expensive matrix multiplications.

In addition to the first-order coherency condition, we will also assume that the coarse model is also *second-order coherent*,

$$\mathbf{R}\nabla f^2(\mathbf{x}_k)\mathbf{P} = \nabla^2 F(\mathbf{y_0}). \tag{4}$$

A simple and surprisingly effective method to construct the coarse model in equation 2 while satisfying both the first and second order coherency conditions in equation 3 and equation 4 is the so-called *Galerkin model*,

$$F(\mathbf{y}) := \langle \mathbf{R}\nabla f(\mathbf{x}_k), \mathbf{y} - \mathbf{y}_0 \rangle + \frac{1}{2}\langle \mathbf{R}\nabla^2 f(\mathbf{x}_k)\mathbf{P}(\mathbf{y} - \mathbf{y}_0), \mathbf{y} - \mathbf{y}_0 \rangle. \tag{5}$$

In the context of multilevel optimization methods, the Galerkin model was experimentally tested in Gratton et al. (2010) where the authors found that it compared favorably with other methods to construct coarse-grained models. Motivated by the excellent numerical results in Gratton et al. (2010), and its simplicity, we will adopt the model in equation 5 as the coarse-grained model in the proposed algorithm. The model in equation 5 also has close links with the recently proposed randomized Newton methods which we briefly discuss below (for more details see Tsipinakis & Parpas (2024); Ho et al. (2019)).

We compute the *coarse direction* by $\hat{\mathbf{d}}_{H,k} := \mathbf{y}^\star - \mathbf{y}_0$. Furthermore, in the case of the Galerkin model defined in equation 5 we have the closed form solution,

$$\hat{\mathbf{d}}_{H,k} = \arg\min_{\mathbf{d}_H \in \mathbb{R}^N} \left\{ \frac{1}{2}\|[\nabla^2 F(\mathbf{y_0})]^{1/2}\mathbf{d}_H\|_2^2 + \langle \nabla F(\mathbf{y}_0), \mathbf{d}_H \rangle \right\} = -[\nabla^2 F(\mathbf{y_0})]^{-1}\nabla F(\mathbf{y_0}). \tag{6}$$

Note that the coarse direction is a vector in $\mathbb{R}^N$. To correct the incumbent solution $\mathbf{x}_k$ we must prolongate it to $\mathbb{R}^n$ using the prolongation operator,

$$\hat{\mathbf{d}}_{h,k} := \mathbf{P}\hat{\mathbf{d}}_{H,k} = -\mathbf{P}[\nabla^2 F(\mathbf{y_0})]^{-1}\nabla F(\mathbf{y_0}), \tag{7}$$

where the $h$ and $H$ subscripts denote directions related to the fine and coarse model respectively. If we naively set $\mathbf{P}$ as the identity matrix, the search direction in equation 7 becomes the Newton direction. When $\mathbf{P}$ is selected as in Definition 1 then we obtain the randomized Newton method which is based on uniform sampling. In this case, the cost to compute the reduced Hessian is $\mathcal{O}(nN)$ which is much cheaper than $\mathcal{O}(n^2)$ of the Newton method (for details on how to compute the reduced Hessian matrix see Remark 2.1 in Tsipinakis & Parpas (2024)). The general multilevel method of this section achieves a local super-linear convergence for both self-concordant and strongly convex functions Tsipinakis & Parpas (2024); Ho et al. (2019). However, when assuming self-concordant function, one can additionally show that the general multilevel method enjoys a global and scale invariant convergence analysis (Tsipinakis & Parpas, 2024). In the next section, we develop a variant of the general multilevel method which achieves a global analysis with a local super-linear convergence rate, which is also applicable to non-convex functions.

## 3  Low-rank Multilevel Newton Methods

We saw above that the general multilevel method can be viewed as a randomized Newton method. In this section we discuss connections of the multilevel scheme with the low-rank Newton method through the naive Nyström method and propose constructing the coarse model using a Truncated Singular Value Decomposition (T-SVD). The version of our method which is suitable for non-convex optimization problems is given in Algorithm 1.

Let $\mathbf{A} \in \mathbb{R}^{n \times n}$ be a positive definite matrix and $\mathbf{Y} \in \mathbb{R}^{n \times N}$ with $\text{rank}(\mathbf{Y}) = N < n$. The Nyström method builds a rank-$N$ approximation of $\mathbf{A}$, namely $\mathbf{A}_N$, as follows,

$$\mathbf{A}_N := \mathbf{AY}(\mathbf{Y^T A Y})^{-1}(\mathbf{AY})^T \approx \mathbf{A}. \tag{8}$$

The Nyström method has been extensively studied and has been shown to be an efficient low-rank approximation method for different sampling techniques which gives practitioners the freedom to choose from a wide range of random matrices (for more details see Drineas & Mahoney (2005); Gittens (2011); Smola & Schölkopf (2000); Williams & Seeger (2000)). Then, one may obtain the naive Nyström method when $\mathbf{Y}$ is selected as in Definition 1. Similarly, we may obtain a rank-$N$ approximation of $\mathbf{A}$ by T-SVD,

$$\mathbf{A}_N := \mathbf{U}_N \Sigma_N \mathbf{U}_N^T.$$

For $\mathbf{A}$ being a symmetric positive definite matrix, $\mathbf{\Sigma}_N \in \mathbb{R}^{N \times N}$ is a diagonal matrix containing the $N$-largest eigenvalues of $\mathbf{A}$, with $\sigma_1 \geq \sigma_2 \geq \cdots \geq \sigma_N$, and $\mathbf{U}_N \in \mathbb{R}^{n \times N}$ contains the corresponding eigenvectors. Although the naive Nyström method is more efficient compared to the T-SVD method, for the algorithms we propose below we employ the T-SVD as it gives us direct access to the eigenvalues of the Hessian matrix.

### 3.1  Convergence Analysis of Low-rank Newton Method for Self-concordant functions

Based on the Nyström method in equation 8, substituting $\mathbf{A}$ with $\nabla^2 f(\mathbf{x}_k)$, $\mathbf{Y}$ with $\mathbf{P}$ and multiplying right and left with $[\nabla^2 f(\mathbf{x}_k)]^{-1}$ we obtain,

$$[\nabla^2 f(\mathbf{x}_k)]^{-1} \approx \mathbf{P}(\mathbf{R}\nabla^2 f(\mathbf{x}_k)\mathbf{P})^{-1}\mathbf{R}, \tag{9}$$

which implies that the coarse direction in equation 7 can be viewed as an approximation of the Newton direction that is based on a low-rank approximation of the Hessian matrix. The fact that equation 9 offers a low rank approximation of the Hessian matrix gives us the freedom to design search directions using different low-rank approximation approaches. Here, similar to the approach proposed in Erdogdu & Montanari (2015), we perform a T-SVD approximation method (note that the work of Erdogdu & Montanari (2015) is based on sub-sampling and thus it is different to ours and not directly comparable). In particular, we construct an approximation of the inverse Hessian, namely $\mathbf{Q}_{h,k}^{-1}$, by computing the $(N+1)^{\text{th}}$ T-SVD of $\nabla^2 f(\mathbf{x}_k)$ and then replace all the eigenvalues after the $(N+1)^{\text{th}}$ eigenvalue with $\sigma_{N+1}$. Formally, $\mathbf{Q}_{h,k}^{-1}$ is constructed as follows:

$$\mathbf{Q}_{h,k}^{-1} := \sigma_{k,N+1}^{-1}\mathbf{I}_{n \times n} + \mathbf{U}_{k,N}(\mathbf{\Sigma}_{k,N}^{-1} - \sigma_{k,N+1}^{-1}\mathbf{I}_{N \times N})\mathbf{U}_{k,N}^T, \tag{10}$$

where $\mathbf{U}_{k,N}$ and $\mathbf{\Sigma}_{k,N}$ are the matrices of eigenvectors and eigenvalues obtained by the T-SVD process at $\mathbf{x}_k$, respectively. Thus, by definition, the eigenvalues of $\mathbf{Q}_{h,k}^{-1}$ have the form $\text{diag}\left(1/\sigma_{k,1}, \ldots, 1/\sigma_{k,N}, 1/\sigma_{k,N+1}, \ldots, 1/\sigma_{k,N+1}\right)$. Then, $\hat{\mathbf{d}}_{h,k} = -\mathbf{Q}_{h,k}^{-1}\nabla f(\mathbf{x}_k)$ is a descent direction. As a result, for convex (stronlgy or self-concordant) problems, this approach is efficient when the valuable second-order information is concentrated on the first, say $N$, eigenvalues. In particular, eigenvalues of large magnitude correspond to more informative directions since in these directions the objective function has a large curvature, whereas for eigenvalues that are close to zero the curvature becomes almost flat. Therefore, by employing equation 10 in the computation of the search direction equation 7 we aim to determine the subspace spanned by the dominant eigenvectors associated with the largest eigenvalues, which are those that yield the fast convergence rate of the Newton method. Such problem structures are typical in machine learning problem (Berahas et al., 2017), e.g., the Hessian matrix is (nearly) low-rank and/or there is a big gap

between the $\sigma_N$ and $\sigma_{N+1}$ eigenvalues. In the convergence analysis, we make use of the Newton decrement, which is given by,

$$\lambda(\mathbf{x}_k) := \left[\nabla f(\mathbf{x}_k)^T \nabla^2 f(\mathbf{x}_k)^{-1} \nabla f(\mathbf{x}_k)\right]^{1/2}.$$

We also define the following quantities

$$\varepsilon_k := \frac{\sigma_{k,n}}{\sigma_{k,N+1}}, \quad \varepsilon := \liminf_{k \to \infty} \varepsilon_k, \quad \varepsilon_{\min} := \inf\{\varepsilon_k \mid k \in \mathbb{N}\}.$$

We are now in position to present the convergence analysis of the method. The convergence is split into two phases according to the magnitude of $\lambda(\mathbf{x}_k)$. If $\lambda(\mathbf{x}_k)$ is sufficiently small, then the method enters the fast region of convergence. The fast region is governed by $\eta := (3 - \sqrt{9 - 4\varepsilon_{\min}})/2$. As the theorem shows, the rate of convergence depends on the magnitude of $\varepsilon$.

**Theorem 3.1.** *Let $f$ be a strictly convex self-concordant function and suppose that the sequence $(\mathbf{x}_k)_{k \in \mathbb{N}}$ is generated by $\mathbf{x}_{k+1} = \mathbf{x}_k - t_k \mathbf{Q}_{h,k}^{-1} \nabla f(\mathbf{x}_k)$, where $\mathbf{Q}_{h,k}^{-1}$ as in (10). Suppose also that $\varepsilon \neq 0$. Then, there exist $\gamma > 0$ and $\eta \in (0, \frac{3-\sqrt{5}}{2})$ such that*

*(i) if $\lambda(\mathbf{x}_k) > \eta$, then $f(\mathbf{x}_{k+1}) - f(\mathbf{x}_k) \leq -\gamma$*

*(ii) if $\lambda(\mathbf{x}_k) \leq \eta$, then the line search selects the unit step and*

*(a) if $\varepsilon \in (0, 1)$, then the method achieves a composite convergence rate, i.e.,*

$$\lambda(\mathbf{x}_{k+1}) \leq \frac{1 - \varepsilon_{\min} + \lambda(\mathbf{x}_k)}{(1 - \lambda(\mathbf{x}_k))^2} \lambda(\mathbf{x}_k) < \lambda(\mathbf{x}_k)$$

*(b) if $\varepsilon = 1$, then the method converges with at least super-linear rate, i.e.,*

$$\frac{\lambda(\mathbf{x}_{k+1})}{\lambda(\mathbf{x}_k)} \leq \frac{1 - \varepsilon_k + \lambda(\mathbf{x}_k)}{(1 - \lambda(\mathbf{x}_k))^2}$$

The proof of the theorem appears in Appendix B. The convergence rate depends on the value of $\varepsilon$. If $\varepsilon \in (0, 1)$, then the method attains a composite rate. If $\varepsilon = 1$, then the sequence $(\varepsilon_k)_{k \in \mathbb{N}}$ converges to one since $\limsup_{k \to \infty} \varepsilon_k = 1$. In this case the method converges at least super-linearly. Quadratic convergence rate will be achieved if $\varepsilon = 1$ and there exists some $k_0 \in \mathbb{N}$ such that $\varepsilon_k = 1$ for all $k \geq k_0$. The theorem does not discuss explicitly the case $\varepsilon = 0$. However, it is readily seen that if this scenario occurs, then $\eta = 0$ and thus the method converges as in phase (i). Furthermore, unlike the classical theory for strongly convex functions, the advantage of using self-concordance as our main assumption results in obtaining a global analysis that has an intuitive local fast convergence rate, which only depends on the ratio between $\sigma_n$ and $\sigma_{N+1}$. The cost per iteration of the proposed method is $\mathcal{O}(Nn^2)$. This is much better than $\mathcal{O}(n^3)$ of the full Newton method.

### 3.1.1 Extension to Non-convex Problems

For self-concordant and other convex functions finding $\mathbf{x}^*$ can in many instances be considered as an easy task (e.g., when descent methods are applicable) since all local minima are global. In this case, the unique global minimum can be attained at an $\mathbf{x}$ for which $\|\nabla f(\mathbf{x})\| = 0$. On the other hand, for non-convex problems, finding $\mathbf{x}^*$ is in general an NP hard problem. For the latter class of problems, a point $\mathbf{x}$ for which $\|\nabla f(\mathbf{x})\| = 0$ can be one of the many local minima or a critical point such as a saddle, for which the Hessian matrix is indefinite. Here, we are concerned with the task of finding a local minimum of a general possible non-convex function $f$. To achieve this, we propose a variant of the low-rank Newton method which we conjecture will have a better escape rate from saddles compared to (stochastic) first-order methods (our conjecture is validated by numerical experiments).

Constructing the coarse direction using the definition in equation 10 is particularly suitable for convex problems since all eigenvalues are positive. Then, for any $i \in \{1, 2, \ldots, n-1\}$ we have $\sigma_i > 0$ which ensures the descent property of $\hat{\mathbf{d}}_{h,k}$. However, when dealing with non-convex functions such guarantee may not

---

**Algorithm 1** SigmaSVD

---

1: Input: $p < N$, $\nu \in (0,1)$, $\epsilon \in (0, 0.68^2)$, $\alpha \in (0, 0.5)$, $\beta \in (0,1)$, $\mathbf{P}_k \in \mathbb{R}^{n \times N}$, $\mathbf{x}_{h,0} \in \mathbb{R}^n$

2: Compute $|\mathbf{Q}_{H,k}^{-1}|$ by equation 16 using the randomized T-SVD (Halko et al., 2011)

3: Form $|\hat{\mathbf{d}}_{h,k}|$ by equation 17

4: Quit if $-\langle \nabla f_{h,k}(\mathbf{x}_{h,k}), |\hat{\mathbf{d}}_{h,k}| \rangle \leq \epsilon$

5: Armijo search: while $f_h(\mathbf{x}_{h,k} + t_k|\hat{\mathbf{d}}_{h,k}|) > f_h(\mathbf{x}_{h,k}) + \alpha t_{h,k} \nabla f_{h,k}^T(\mathbf{x}_{h,k})|\hat{\mathbf{d}}_{h,k}|$, $t_{h,k} \leftarrow \beta t_{h,k}$

6: Update: $\mathbf{x}_{h,k+1} := \mathbf{x}_{h,k} + t_{h,k}|\hat{\mathbf{d}}_{h,k}|$, go to 2

7: Return $\mathbf{x}_{h,k}$

---

be true, e.g., at a neighborhood of a saddle point. Since for the Newton method the negative gradient is pre-multiplied by the inverse Hessian matrix, a negative eigenvalue will result in changing the sign of the corresponding gradient entry which may yield the Newton method converging to a saddle point or a local maximum. On the other hand, the Newton method breaks down when there are zero eigenvalues. To address these shortcomings, we compute the $(N+1)^{\text{th}}$ T-SVD of the Hessian as above, but here we replace all the negative eigenvalues by their absolute value. Further, sufficiently small eigenvalues are replaced by a positive scalar to ensure the non-singularity of the approximated Hessian matrix. Formally, we define $g_{k,i} : \mathbb{R}^{N \times N} \to \mathbb{R}, i = 1, 2, \ldots, N$, such that,

$$g_{k,i}([\boldsymbol{\Sigma}_{k,N}]) := \max\{|[\boldsymbol{\Sigma}_{k,N}]_{ii}|, \nu\}, \ \nu > 0, \tag{11}$$

and $g_k(\boldsymbol{\Sigma}_{k,N}) := \mathrm{diag}(g_{k,1}([\boldsymbol{\Sigma}_{k,N}]), \ldots, g_{k,N}([\boldsymbol{\Sigma}_{k,N}]))$. Then we obtain the truncated low-rank approximation of the Hessian matrix as follows,

$$|\mathbf{Q}_{h,k}^{-1}| := g_k(\sigma_{k,N+1})^{-1}\mathbf{I}_{n \times n} + \mathbf{U}_{k,N}(g_k(\boldsymbol{\Sigma}_{k,N})^{-1} - g_k(\sigma_{k,N+1})^{-1}\mathbf{I}_{N \times N})\mathbf{U}_N^T. \tag{12}$$

Defining the approximated inverse Hessian matrix as above is necessary in order to obtain a descent direction, which together with an appropriately selected step-size parameter we can guarantee the descent nature of the Low-Rank Newton method for non-convex problems. Despite the lower iteration cost compared to the Newton method, there still may be cases that forming the Hessian matrix and computing the T-SVD is too expensive for some applications (when $n$ is extremely large). In the next section, we address this issue.

## 3.2 Coarse-grained Low-rank Newton Method with Analysis for Self-concordant Functions

The computational bottleneck of the procedure described in the previous section arises from the fact that the computations are performed over the full Hessian. To address this, we perform the T-SVD on the Hessian matrix of the reduced order model. We begin by describing a low-rank multilevel method which is suitable for convex optimization. Later, we will present the extension for non-convex functions.

Let $\mathbf{Q}_{H,k}$ be a T-SVD-based rank-$p$ approximation of the reduced Hessian matrix in equation 4, where $p < N < n$. Then, as in equation 9, we take

$$\hat{\mathbf{Q}}_{h,k}^{-1} := \mathbf{P}\mathbf{Q}_{H,k}^{-1}\mathbf{R} \approx \mathbf{P}[\mathbf{R}\nabla^2 f(\mathbf{x}_k)\mathbf{P}]^{-1}\mathbf{R} = \mathbf{Q}_{h,k}^{-1} \approx [\nabla^2 f(\mathbf{x}_k)]^{-1},$$

which in fact is a rank-$p$ approximation of the inverse Hessian matrix with the difference that the computational cost of forming $\hat{\mathbf{Q}}_{h,k}^{-1}$ is significantly reduced since we compute $[\mathbf{U}_{k,p+1}, \boldsymbol{\Sigma}_{k,p+1}]$ (by T-SVD) over the reduced Hessian matrix equation 4. We wish our Hessian approximation to have a similar structure as in equation 10, thus, we define

$$\mathbf{Q}_{H,k}^{-1} := \sigma_{k,p+1}^{-1}\mathbf{I}_{N \times N} + \mathbf{U}_{k,p}(\boldsymbol{\Sigma}_{k,p}^{-1} - \sigma_{k,p+1}^{-1}\mathbf{I}_{p \times p})\mathbf{U}_{k,p}^T, \tag{13}$$

where $\mathbf{U}_{k,p} \in \mathbb{R}^{N \times p}$. Therefore, $\hat{\mathbf{d}}_{h,k} = -\hat{\mathbf{Q}}_{h,k}^{-1}\nabla f(\mathbf{x}_k)$ is an approximation of the Newton direction. We call this method SigmaSVD as it is an approximation of Sigma in Tsipinakis & Parpas (2024). Let us now define the approximate decrements for both SigmaSVD and the general multilevel method of section 2,

$$\hat{\lambda}(\mathbf{x}_k) := \left[\nabla f(\mathbf{x}_k)^T \hat{\mathbf{Q}}_{h,k}^{-1} \nabla f(\mathbf{x}_k)\right]^{1/2},$$

$$\tilde{\lambda}(\mathbf{x}_k) := \left[(\mathbf{R}\nabla f(\mathbf{x}_k))^T [\nabla^2 F(\mathbf{y}_0)]^{-1} \mathbf{R}\nabla f(\mathbf{x}_k)\right]^{1/2}, \tag{14}$$

respectively. Both quantities are analogous to the Newton decrement $\lambda(\mathbf{x}_k)$ and serve the same purpose. In this section, we will construct the operators randomly at each iterations according to Definition 1, thus we slightly change the notation of random operators to $\mathbf{P}_k$ and $\mathbf{R}_k$. This way we enhance the applicability of the method. The following assumption emerges naturally.

**Assumption 2.** *For all $k \in \mathbb{N}$ it holds $\hat{\lambda}(\mathbf{x}_k) > 0$ with some probability $\delta > 0$.*

The above assumption is typical when analyzing probabilistic subspace methods (Cartis et al., 2022; Tsipinakis & Parpas, 2024; Ho et al., 2019). Since $\hat{\lambda}(\mathbf{x}_k)$ is a norm of $\mathbf{R}_k \nabla f(\mathbf{x}_k)$, assumption 2 effectively prevents the latter from becoming zero with some probability. This assumptions is needed when analyzing these methods because $\mathbf{R}_k \nabla f(\mathbf{x}_k) = 0$ can occur when $\mathbf{R}_k \in \text{null}(\nabla f(\mathbf{x}_k))$.

When assuming self-concordant functions, the convergence analysis of the proposed method is similar to that in Tsipinakis & Parpas (2024), with the difference that the term that controls its convergence rate is given by $\hat{e}_k := (\lambda^2(\mathbf{x}_k) - \hat{\lambda}^2(\mathbf{x}_k))^{1/2}$ instead of $\tilde{e}_k := (\lambda^2(\mathbf{x}_k) - \tilde{\lambda}^2(\mathbf{x}_k))^{1/2}$. It has been shown that $\tilde{e}_k \leq (1 - \mu_k^2)^{1/2} \lambda(\mathbf{x}_k)$, $0 < \mu_k \leq 1$ (Tsipinakis & Parpas, 2024). Intuitively, it should be expected $\hat{e}_k$ to further incorporate the information carried out by the T-SVD. Self-concordant functions allow us to prove such an informative upper bound.

**Lemma 3.1.** *For all $k \in \mathbb{N}$ we have that*

1. $\mathbf{d}_{h,k}^T \nabla^2 f(\mathbf{x}_k) \hat{\mathbf{d}}_{h,k} = \hat{\lambda}^2(\mathbf{x}_k)$

2. $\hat{\mathbf{d}}_{h,k}^T \nabla^2 f(\mathbf{x}_k) \hat{\mathbf{d}}_{h,k} \leq \hat{\lambda}^2(\mathbf{x}_k)$

3. $\|[\nabla^2 f(\mathbf{x}_k)]^{\frac{1}{2}} (\mathbf{d}_{h,k} - \hat{\mathbf{d}}_{h,k})\| \leq \hat{e}_k$

4. $\sqrt{\frac{\sigma_{k,N}}{\sigma_{k,p+1}}} \tilde{\lambda}(\mathbf{x}_k) \leq \hat{\lambda}(\mathbf{x}_k) \leq \tilde{\lambda}(\mathbf{x}_k) \leq \lambda(\mathbf{x}_k)$

*Further, let Assumption 2 hold. Then, there exists $\mu_k \in (0,1]$ such that $\hat{e}_k \leq (1 - \frac{\sigma_{k,N}}{\sigma_{k,p+1}} \mu_k^2)^{1/2} \lambda(\mathbf{x}_k)$ with probability $\delta$.*

The proof of the theorem below is based on the proof technique of Theorem 3.1. We define the following quantities

$$\bar{\varepsilon}_k := \frac{\sigma_{k,N}}{\sigma_{k,p+1}}, \ \bar{\varepsilon}_{\min} := \inf\{\varepsilon_k \mid k \in \mathbb{N}\}, \ \bar{\varepsilon} := \liminf_{k \to \infty} \varepsilon_k,$$
$$\mu := \liminf_{k \to \infty} \mu_k, \ \mu_{\min} := \inf\{\mu_k \mid k \in \mathbb{N}\} \tag{15}$$

We will show that the region of the fast convergence is governed by $\hat{\eta} := \frac{3 - \sqrt{5 + 4\hat{\varepsilon}}}{2}$, where $\hat{\varepsilon} := \sqrt{1 - \bar{\varepsilon}_{\min} \mu_{\min}^2}$.

**Theorem 3.2.** *Suppose that $f$ is a strictly self-concordant function, $\mathbf{P}_k$ is selected as in Definition 1 and $\bar{\varepsilon}_{\min}, \mu_{\min} \neq 0$. Suppose also that Assumption 2 holds and the sequence $(\mathbf{x}_k)_{k \in \mathbb{N}}$ is generated by $\mathbf{x}_{k+1} = \mathbf{x}_k - t_k \hat{\mathbf{Q}}_{h,k}^{-1} \nabla f(\mathbf{x}_k)$. Then, there exist constants $\hat{\gamma} > 0$ and $\hat{\eta} \in (0, \frac{3-\sqrt{5}}{2})$ such that*

*(i) if $\lambda(\mathbf{x}_{h,k}) > \hat{\eta}$, then*

$$f_h(\mathbf{x}_{h,k+1}) - f_h(\mathbf{x}_{h,k}) \leq -\hat{\gamma},$$

*with probability $\delta$ at each $k \in \mathbb{N}$,*

*(ii) if $\lambda(\mathbf{x}_{h,k}) \leq \hat{\eta}$, and*

*(a) if either $\bar{\varepsilon} \in (0,1)$ or $\mu \in (0,1)$, then*

$$\lambda(\mathbf{x}_{k+1}) \leq \frac{\sqrt{1 - \bar{\varepsilon}_{\min} \mu_{\min}^2} + \lambda(\mathbf{x}_k)}{(1 - \lambda(\mathbf{x}_k))^2} \lambda(\mathbf{x}_k) < \lambda(\mathbf{x}_k),$$

*and thus the sequence $(\lambda(\mathbf{x}_k))_{k \in \mathbb{N}}$ achieves a composite rate with probability $\delta$ at each $k \in \mathbb{N}$,*

*(b) if $\bar{\varepsilon} = \mu = 1$, then*

$$\frac{\lambda(\mathbf{x}_{k+1})}{\lambda(\mathbf{x}_k)} \leq \frac{\sqrt{1 - \bar{\varepsilon}_k \mu_k^2} + \lambda(\mathbf{x}_k)}{(1 - \lambda(\mathbf{x}_k))^2},$$

*and thus the sequence $(\lambda(\mathbf{x}_k))_{k \in \mathbb{N}}$ converges super-linearly with probability $\delta$ at each $k \in \mathbb{N}$.*

The proof of the theorem appears in Appendix C. The difference between theorem 3.1 and 3.2 is that the latter involves $\mu_k$, which effectively is the quantity that controls the level of approximation between the coarse and fine directions. Therefore, a slower convergence rate is expected compared to the low-rank Newton method in section 3.1. If $\mu_k$ approaches one, then SigmaSVD will achieve the fast rate of the low-rank Newton method. The method is also an approximation of the multilevel method in Tsipinakis & Parpas (2024). Theorem 3.2 shows that SigmaSVD will converge similar to the standard multilevel method in Tsipinakis & Parpas (2024) if $\bar{\varepsilon}_k$ is large enough. As in theorem 3.1, if $\bar{\varepsilon} = 0$ or $\mu = 0$, then $\hat{\eta} = 0$ and the method will converge as in phase (i).

### 3.3 Analysis for Non-convex Functions and Polyak-Lojasiewicz Inequality

In this section we analyze SigmaSVD abandoning self-concordance and convexity from our assumptions. Specifically, assuming Lipschitz continuous gradients, we show reduction in the value of the objective function for sequences $(\mathbf{x}_k)_{k \in \mathbb{N}}$ generated using the truncated corse direction equation 17 below. If we further assume that our objective function satisfies the Polyak-Lojasiewicz (PL) inequality, then the proposed method will converge with a linear rate.

Based on the discussion in section 3.1.1 and given the eigenvalue decomposition which, as in 3.2, returns $\mathbf{\Sigma}_{k,p+1}$ and $\mathbf{U}_{k,p+1}$, the truncated reduced Hessian matrix is defined as follows

$$|\mathbf{Q}_{H,k}^{-1}| := g_{k,p+1}(\mathbf{\Sigma}_{k,p+1})^{-1}\mathbf{I}_{N \times N} + \mathbf{U}_{k,p}(g_k(\mathbf{\Sigma}_p)^{-1} - g_{k,p+1}(\mathbf{\Sigma}_{k,p+1})^{-1}\mathbf{I}_{p \times p})\mathbf{U}_{k,p}^T, \tag{16}$$

then the truncated coarse direction is given by

$$|\hat{\mathbf{d}}_{h,k}| := -|\hat{\mathbf{Q}}_{h,k}^{-1}|\nabla f(\mathbf{x}_k) = -\mathbf{P}|\mathbf{Q}_{H,k}^{-1}|\mathbf{R}\nabla f(\mathbf{x}_k). \tag{17}$$

The full algorithm together with a step-size strategy is specified in Algorithm 1. When the objective function is convex or self-concordant Algorithm 1 coincides with SigmaSVD in section 3.2.

**Assumption 3.** *The gradient of $f$ is Lipschitz continuous on $\mathbb{R}^n$, that is, there exists $M > 0$ such that for all $\mathbf{x}, \mathbf{y} \in \mathbb{R}^n$*

$$f(\mathbf{y}) \leq f(\mathbf{x}) + \nabla f(\mathbf{x})^T(\mathbf{y} - \mathbf{x}) + \frac{M}{2}\|\mathbf{y} - \mathbf{x}\|^2,$$

A direct consequence of the above assumption is the boundness of the Hessian matrix from above: $\nabla^2 f(\mathbf{x}) \preceq M\mathbf{I}_N$, for all $\mathbf{x} \in \mathbb{R}^n$.

**Assumption 4.** *There exists $\hat{\mu} \in (0, 1]$ such that $\|\mathbf{R}_k \nabla f(\mathbf{x}_k)\| \geq \hat{\mu}\|\nabla f(\mathbf{x}_k)\|$ with probability $\hat{\delta} > 0$.*

Assumption 4 is typical when analyzing subspace methods, see Cartis et al. (2022); Cartis & Roberts (2024); Cartis et al. (2025). In this section we use it in place of assumption 2 to prevent ineffective coarse steps with probability $\hat{\delta}$.

**Theorem 3.3.** *Let assumptions 3 and 4 hold. Suppose also that the sequence $(\mathbf{x}_k)_{k \in \mathbb{N}}$ is generated by $\mathbf{x}_{k+1} = \mathbf{x}_k + t_k|\hat{\mathbf{d}}_{h,k}|$ and $t_k \leq \frac{|\sigma_{k,p+1}|}{M\omega^2}$. Then*

$$f(\mathbf{x}_{k+1}) - f(\mathbf{x}_k) \leq -\frac{\hat{\mu}^2 \nu}{2\omega^2 M}\|\nabla f(\mathbf{x}_k)\|^2,$$

*with probability $\delta$.*

The above result shows that we can obtain a descent method when the function has Lipschitz gradients. The result is global and remains true with probability $\delta$. The method will not diverge since, with probability $1-\delta$, $\hat{\mu}$ may be zero, and thus $f(\mathbf{x}_{k+1}) = f(\mathbf{x}_k)$. However, due to the randomness of the method, at a future iteration, it is expected that the coarse model will not be ineffective (and thus $\hat{\mu} \neq 0$), which effectively yields a convergent algorithm. In order to derive a rate of convergence, we make use of the following assumption.

**Assumption 5.** *The PL inequality holds; there exists $\xi > 0$ such that*

$$\frac{1}{2}\|\nabla f(\mathbf{x}_k)\|^2 \geq \xi(f(\mathbf{x}_k) - f(\mathbf{x}^*)). \tag{18}$$

The PL condition is weaker than convexity (see Karimi et al. (2016) for more details) and has been (empirically) shown that it is often satisfied for over-parameterized neural networks (see Belkin (2021) and references therein). We think that this result goes some way to explaining the behavior of the algorithm (at least close enough to a minimum).

**Theorem 3.4.** *Let assumptions 3, 5 and 4 hold. Assume that $(\mathbf{x}_k)_{k\in\mathbb{N}}$ is generated by $\mathbf{x}_{k+1} = \mathbf{x}_k + \frac{|\sigma_{k,p+1}|}{M\omega^2}|\hat{\mathbf{d}}_{h,k}|$. Then*

$$f(\mathbf{x}_{k+1}) - f(\mathbf{x}^*) \leq \left(1 - \frac{\nu\xi\hat{\mu}^2}{2\omega^2 M^2}\right)(f(\mathbf{x}_k) - f(\mathbf{x}^*))$$

*with probability $\hat{\delta}$.*

Refer to Appendix D for the proof. The theorem presented is global, demonstrating that convergence to $f(\mathbf{x}^*)$ can be achieved with at least a linear rate. Theorems 3.3 and 3.4 become particularly effective when both $\hat{\mu}$ and $\delta$ are sufficiently large. Objective functions with high $\hat{\mu}$ and $\delta$ often include those with low-rank Hessians or concentrated second-order information in the dominant eigenvalues. Functions satisfying the PL inequality typically exhibit low-rank Hessians. Given the method's low per-iteration computational cost ($\mathcal{O}(N + pN^2)$ in parallel), algorithm 1 is expected to escape saddle points efficiently and converge rapidly to a local minimum. This will be empirically validated in the following section.

**Remark 3.1.** *Since the Euclidean norm is used to obtain the convergence results in this section, we are in a position to quantify the parameters $\hat{\mu}$ and $\hat{\delta}$ depending on the choice of the operator $\mathbf{R}$. If $\mathbf{R}$ is a Gaussian matrix whose entries are drawn from $N(0, N^{-1})$, then for any $\hat{\mu} \in (0,1)$ we have $\hat{\delta} = e^{-(1-\hat{\mu}^2)^2 N/4}$ (Cartis et al., 2022). Another possible choice that yields sparse matrices, and is therefore less computationally expensive than using dense Gaussian matrices, is given by s-hashing matrices (Kane & Nelson, 2014). When using s-hashing matrices to construct the coarse models, then for any $\hat{\mu} \in (0,1)$ we have $\hat{\delta} = e^{-N(1-\hat{\mu}^2)^2/C_1}$, where $s = C_2(1 - \hat{\mu}^2)N$, and $C_1, C_2$ are problem-dependent constants. To the best of our knowledge, such explicit quantitative values for $\hat{\mu}$ and $\hat{\delta}$ under definition 1 are not available in the literature.*

**Remark 3.2.** *In practice, we select $N$ and $p$ as follows: When $\mathbf{R}$ is constructed as in definition 1, these parameters should be selected according to the effective rank of the Hessian at the optimum, $R = \frac{\text{tr}(\nabla^2 f(\mathbf{x}^*))}{\lambda_{\max}(\nabla^2 f(\mathbf{x}^*))}$. The quantity $R$ indicates how many directions really matter near the solution, where most methods slow down significantly. If $R$ is known, in practice $N$ and $p$ should be selected larger and slightly smaller than $R$, respectively. Alternatively, they can be selected in a similar fashion according to the quantity $\max\{n, R\ln n\}$, as suggested in Erdogdu & Montanari (2015). On the other hand, if $\mathbf{R}$ is a Gaussian or s-hashing $N$ and $p$ should be selected according to Remark 3.1, i.e., by specifying $\hat{\mu}$ and $\hat{\delta}$ one obtains $N$ directly.*

# 4 Numerical results

We are now ready to validate the efficiency of the proposed method on different machine learning models. We illustrate that the method is efficient and compares favorably with other state-of-the-art algorithms on a broad class of problems. In particular, we apply algorithm 1 on problems minimizing *self-concordant*

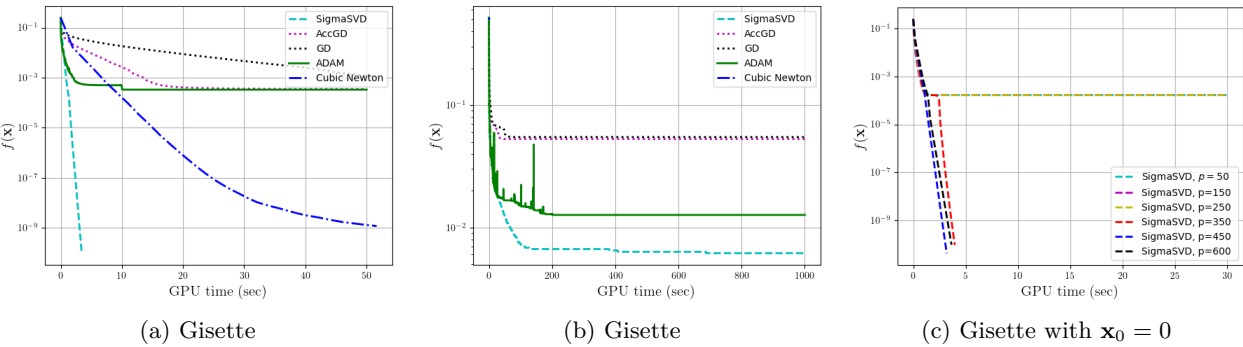

(a) Gisette        (b) Gisette        (c) Gisette with $\mathbf{x}_0 = 0$

Figure 1: Non-convex minimization. All the methods in plot (a) are initialized at the origin, while in plot (b) the initializer is selected randomly by $\mathcal{N}(0, 1)$. Plot (c) shows the convergence behavior of SigmaSVD for different values of $p$.

functions, *strongly convex* functions with or without Lipschitz continuity and a *non-convex* functions. In this section we only report the results for the non-convex cases. The additional numerical results together with a detailed description of the algorithms, training datasets, and the setup used to obtain the results appear in the Appendix E.

## 4.1 Non-linear least-squares

Suppose that we are given a training set $\{\mathbf{a}_i, b_i\}_{i=1}^m$ with $\mathbf{a}_i \in \mathbb{R}^n$ and $b_i \in [0, 1]$ and consider solving the non-linear least-squares problem,

$$\min_{\mathbf{x} \in \mathbb{R}^n} \frac{1}{m} \sum_{i=1}^m (b_i - \phi(\mathbf{a}_i^T \mathbf{x}))^2, \quad \phi \ : \ w \to \frac{1}{1 + \exp(w)^{-1}},$$

where $\phi$ is a real function, known as sigmoid. Since we apply the mean squared error over the sigmoid function, the non-linear least-squares is a non-convex problem. We compare the performance of algorithm 1 against gradient descent (GD), accelerated gradient descent (AGD), cubic Newton and Adam. Although Adam, as a batch algorithm, is not directly comparable to SigmaSVD, we include it in our comparisons to demonstrate the advantages of second-order methods compared to the stochastic first-order method on problems with several saddle points and flat areas. For algorithm 1, GD and AGD the Armijo rule is applied to determine the step size parameter with constants $\alpha = 0.001, \beta = 0.7$. A line search strategy is applied to select the regularization parameter of the Cubic Newton method (for details see Nesterov et al. (2018)). In all experiments we select the fixed eigenvalue threshold, i.e., $\nu = 10^{-10}$ in equation 11. The momentum parameters for Adam are selected as suggested in Kingma & Ba (2014) while for AGD the momentum is selected as 0.5.

In Figure 1 we demonstrate the performance of the different optimization methods on the non-linear least-squares problem for the Gisette dataset (see also Figure 3 in Appendix E for simulations on different data regimes). In Figures 1a and 1c we illustrate the reduction of the value function over GPU time. Observe that SigmaSVD is able to return a better solution compared to the first-order methods. Clearly, first-order methods are trapped in a flat area (Hessian and gradient are zero) and thus they are stuck far from the local or global minima. This is not a surprise since in such areas the gradient is almost zero and thus first-order methods are unable to progress and although in theory they have been shown to always escape even ill-conditioned saddles, here we see that they require an extremely large amount of iterations which makes them inefficient for practical applications. On the other hand, Cubic Newton is able to escape saddle points in one iteration, however we observed that in many experiments it converges rapidly to early local minima probably due to the lack of randomness. Moreover, as suggested in Theorem 3.2, Figure 1 illustrates that SigmaSVD enjoys a fast local convergence rate and it is faster than its counterparts. It is also faster and has the ability to achieve lower training errors for different initialization points.

Table 1: Probability of successfully escaping from saddle points and convergence to the global minimum for various values of $N$ and fixed $p = 450, \mathbf{x}_0 = 0$. Each row shows the probability over 50 trials.

| Escape Rate Probability - Gisette | |
|---|---|
| N | Probability |
| $0.1n$ | 18% |
| $0.13n$ | 46% |
| $0.26n$ | 52% |
| $0.36n$ | 66% |
| $0.42n$ | 80% |
| $0.46n$ | 92% |

In Table 1 and Figure 1c we demonstrate how the choice of $N$ and $p$ affects the behavior of SigmaSVD around a saddle or flat area. We revisit the experiment of Figure 1b (i.e., $\mathbf{x}_0 = 0$) in which SigmaSVD escapes from such the flat area in one iteration (similar to Cubic Newton). Figure 1c shows the behavior of SigmaSVD for different values of $p$ with fixed $N = 0.5n$. Observe that for very small values of $p$ SigmaSVD is trapped in the same saddle as the first-order, however for $p \geq 350$ it converges to the global minimum. Next, in Table 1 we fix $p = 450$ and vary $N$ to show the escape rate probability from this region, or otherwise the probability of convergence to the global minimum over 50 trials. Similarly to $p$, we see that the escape rate probability is proportional to $N$. Both experiments verify Theorem 3.2 which indicates that for very small values in $p$ and $N$, one should expect SigmaSVD to behave as a first-order method. Furthermore, Table 1 and Figure 1 show that SigmaSVD is able to reach the behavior of the Cubic Newton method (that is, escape from the saddle point in one iteration) using only half of the dimensions of the problem and about 20% of the eigenvalues of the reduced Hessian, which constitutes a significant improvement in total complexity as also depicted in Figure 1a.

## 4.2 MNIST deep autoencoder

In this section we consider training the MNIST deep autoencoder which is based on the work presented in Hinton & Salakhutdinov (2006). The MNIST auto-encoder is considered a benchmark optimization problem because the presence of saddle points along its optimization trajectory poses significant challenges to optimization algorithms, see Reddi et al. (2018) and reference therein. The MNIST auto-encoder consists of an encoder with layers-size $28 \times 28, 1000, 500, 250, 30$ that it is followed by a symmetric decoder and in total has 2.8M parameters. Further, the sigmoid activation function is applied to all layers.

Here, we compare SigmaSVD against Adam. Since the MNIST autoencoder problem is large, to avoid memory issues, we adapt SigmaSVD for batch learning. In addition we use first-order momentum, namely $\beta_1 \in (0,1)$, similar to that of Adam. In particular, our new iterations have the following form

$$\mathbf{x}_{k+1} = \mathbf{x}_k - t_k \mathbf{P} |\mathbf{Q}_{H,k}^{-1}| \mathbf{R}\hat{\mathbf{m}}_{k+1},$$

where

$$\hat{\mathbf{m}}_{k+1} = \frac{\mathbf{m}_{k+1}}{1 - \beta_1^k}, \quad \mathbf{m}_{k+1} = \beta_1 \cdot \mathbf{m}_k + (1 - \beta_1) \cdot \nabla f(\mathbf{x}_k).$$

As a result, SigmaSVD with momentum effectively applies Adam's updates, substituting the Hessian for the second moment of the gradients (which is a diagonal matrix). Our goal is to demonstrate that the Hessian can be essential in deep architectures and that its diagonal approximations may severely slow down optimization algorithms or lead to suboptimal performance in the presence of saddles points and flat areas.

To obtain the results the batch size is set to 128. Furthermore, for Adam, we set the learning rate to 0.0025 (yields the best convergence rate) while keeping the momentum parameters at their default values (Kingma & Ba, 2014). For SigmaSVD, we use two coarse models of different sizes: one with N = 2,800 and $t_k = 0.01$, and another with N = 1,400 and $t_k = 0.02$, where for both we used $\beta_1 = 0.9$. In both cases, we found that setting $\nu = 10^{-8}$ yielded the best performance. Additionally, we use a full eigenvalue decomposition instead of a truncated SVD. This is feasible because the coarse models are small, allowing for efficient computation.

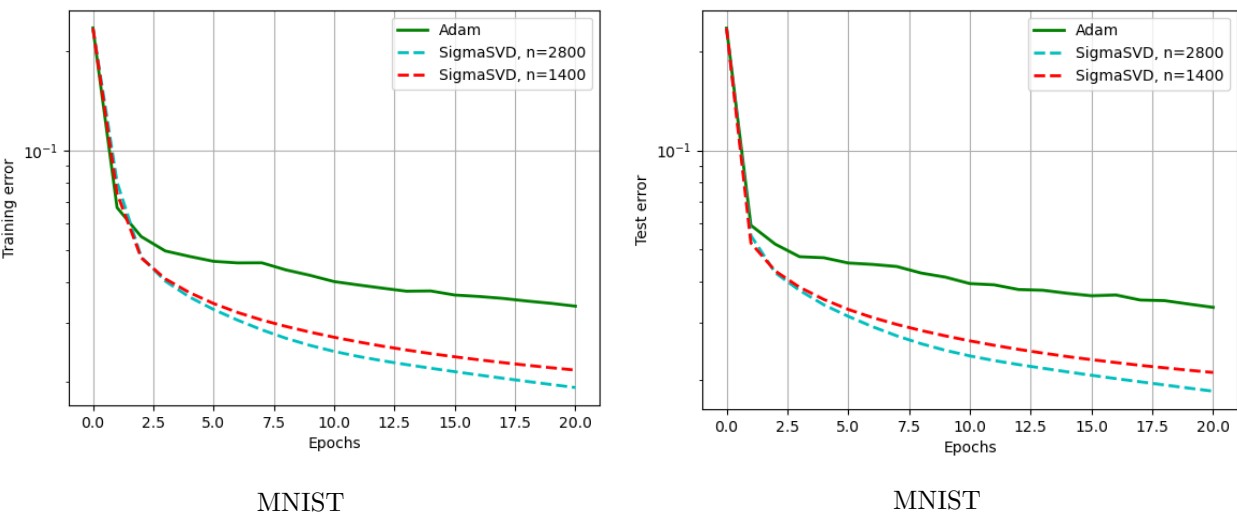

Figure 2: Minimizing the MNIST deep autoencoder. Performance comparison between SigmaSVD and Adam. The left plot shows the reduction in the value of the objective function and the right plot the generalization performance versus epochs.

A full SVD is also advantageous as it eliminates the need to tune an extra hyperparameter ($p$ in this case). More importantly, selecting a larger $p$ generally leads to better escape rates near saddle points, as observed in Figure 1c. We ran the experiments on a single A100 GPU with 40Gb RAM.

The results of this experiment are shown in Figure 2. The left plot illustrates how quickly the algorithms reduce $f(\mathbf{x})$ over epochs. Clearly, both SigmaSVD variants outperform Adam in terms of convergence speed in the first 20 epochs. This significant difference in convergence rate stems from the presence of multiple saddle points and flat regions along the optimization trajectory. In particular, during the early stages of training, we observed instances where the gradient norm was nearly zero (indicating saddle points) and cases where all eigenvalues of the reduced Hessian were zero (suggesting a possible flat region). SigmaSVD handles these situations efficiently, escaping quickly by leveraging second-order information, whereas Adam, relying only on a diagonal approximation, struggles in such ill-conditioned settings. Additionally, the right plot compares the test error, showing that the rapid convergence in training does not translate to overfitting. Additionally, note that the proposed methods achieved these results while updating only $1,400$ and $2,800$ parameters per iteration, respectively, whereas Adam updates all 2.8M parameters. This highlights the significance of employing rich and informative preconditioners, such as the reduced Hessian, in optimizing ill-conditioned deep neural networks. Further, regarding the wall-clock time comparison, Adam completes an epoch significantly faster (18 seconds) compared to the SigmaSVD variants (600 and $1,100$ seconds, respectively). However, comparing convergence based on GPU time would be unfair at this stage, as Adam is a highly optimized industrial algorithm, whereas SigmaSVD has been developed primarily for academic research. Nonetheless, this experiment suggests that, in order to design efficient algorithms for minimizing large neural networks with flat regions and saddle points, one should consider hybrid approaches, combining a computationally inexpensive method such as Adam when gradients are large, with a more sophisticated method such as SigmaSVD near saddle points and flat regions.

## 5    Conclusion

We develop a stochastic multilevel low-rank Newton type method that enjoys a super-linear convergence rate for self-concordant functions with low per-iteration cost. We further propose a variant that is applicable to non-convex problems and establish its linear rate when the PL inequality is satisfied. Preliminary numerical experiments show that our method is efficient for large-scale optimization problems with millions of dimensions. The experiments also show that the method is well-suited to problems with low-rank Hessian matrices.

It is also faster and has an improved escape rate from saddles and flat-areas compared to first-order methods in non-convex cases. As a future direction, we aim to analyze the batch variant of our method and develop a hybrid approach that is efficient when training deep neural networks. We also plan to provide a convergence analysis for general non-convex functions.

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

## A  Background

Throughout this paper, all vectors are denoted with bold lowercase letters, i.e., $\mathbf{x} \in \mathbb{R}^n$ and all matrices with bold uppercase letters, i.e., $\mathbf{A} \in \mathbb{R}^{m \times n}$. The function $\|\mathbf{x}\|_2 = \langle \mathbf{x}^T, \mathbf{x} \rangle^{1/2}$ is the $\ell_2$- or Euclidean norm of $\mathbf{x}$. The spectral norm of $\mathbf{A}$ is the norm induced by the $\ell_2$-norm on $\mathbb{R}^n$ and it is defined as $\|\mathbf{A}\|_2 := \max_{\|\mathbf{x}\|_2=1} \|\mathbf{A}\mathbf{x}\|_2$.

It can be shown that $\|\mathbf{A}\|_2 = \sigma_1(\mathbf{A})$, where $\sigma_1(\mathbf{A})$ (or simply $\sigma_1$) is the largest singular value of $\mathbf{A}$, see (Horn & Johnson, 2013, Section 5.6). For two symmetric matrices $\mathbf{A}$ and $\mathbf{B}$ we write $\mathbf{A} \succeq \mathbf{B}$ when $\mathbf{x}^T(\mathbf{A}-\mathbf{B})\mathbf{x} \geq 0$ for all $\mathbf{x} \in \mathbb{R}^n$, or otherwise when the matrix $\mathbf{A} - \mathbf{B}$ is positive semi-definite. Below we present the main properties and inequalities for self-concordant functions. For a complete analysis see Nesterov (2004); Boyd & Vandenberghe (2004). A univariate convex function $\phi : \mathbb{R} \to \mathbb{R}$ is called self-concordant with constant $M_\phi \geq 0$ if,

$$|\phi'''(x)| \leq M_\phi \phi''(x)^{3/2}. \tag{19}$$

Examples of self-concordant functions include but not limited to linear, convex quadratic, negative logarithmic and negative log-determinant function. Moreover, the class of self-concordant functions includes $\mu$-strongly convex functions with $L$-Lipschitz continuous Hessians, as the latter are self-concordant with constant $M_\phi = L/(2\mu^{3/2})$. Therefore, if $f$ is three-times differentiable, the self-concordance assumption is less restrictive than the classical assumptions, see Nesterov et al. (2018).

Based on eq. (19), a multivariate convex function $f : \mathbb{R}^n \to \mathbb{R}$ is called self-concordant if $\phi(t) := f(\mathbf{x} + t\mathbf{u})$ satisfies equation 19 for all $\mathbf{x} \in \text{dom } f, \mathbf{u} \in \mathbb{R}^n$ and $t \in \mathbb{R}$ such that $\mathbf{x} + t\mathbf{u} \in \text{dom } f$. Further, self-concordance is preserved under composition with any affine function. In addition, for any convex self-concordant function $\tilde{\phi}$ with constant $M_{\tilde{\phi}} \geq 0$ it can be shown that $\phi(x) := \frac{M_{\tilde{\phi}}^2}{4} \tilde{\phi}(x)$ is self-concordant with constant $M_\phi = 2$. Next, given $\mathbf{x} \in \text{dom } f$ and assuming that $\nabla^2 f(\mathbf{x})$ is positive-definite we can define the following norms

$$\|\mathbf{u}\|_{\mathbf{x}} := \langle \nabla^2 f(\mathbf{x})\mathbf{u}, \mathbf{u} \rangle^{1/2} \quad \text{and} \quad \|\mathbf{v}\|_{\mathbf{x}}^* := \langle [\nabla^2 f(\mathbf{x})]^{-1}\mathbf{v}, \mathbf{v} \rangle^{1/2}, \tag{20}$$

for which it holds that $|\langle \mathbf{u}, \mathbf{v} \rangle| \leq \|\mathbf{u}\|_{\mathbf{x}}^* \|\mathbf{v}\|_{\mathbf{x}}$. Therefore, the Newton decrement can be written as

$$\lambda_f(\mathbf{x}) := \|\nabla f(\mathbf{x})\|_{\mathbf{x}}^* = \|[\nabla^2 f(\mathbf{x})]^{-1/2} \nabla f(\mathbf{x})\|_2. \tag{21}$$

In addition, we take into consideration two auxiliary functions, both introduced in Nesterov (2004). Define the functions $\omega$ and $\omega_*$ such that

$$\omega(x) := x - \log(1 + x) \quad \text{and} \quad \omega_*(x) := -x - \log(1 - x), \tag{22}$$

where $\text{dom } \omega = \{x \in \mathbb{R} : x \geq 0\}$ and $\text{dom } \omega_* = \{x \in \mathbb{R} : 0 \leq x < 1\}$, respectively, and $\log(x)$ denotes the natural logarithm of $x$. Moreover, note that both functions are convex and their range is the set of positive real numbers. Further, from the definition equation 19 for $M_\phi = 2$, we have that

$$\left| \frac{d}{dt} \left( \phi''(t)^{-1/2} \right) \right| \leq 1,$$

from which, after integration, we obtain the following bounds

$$\frac{\phi''(0)}{(1 + t\phi''(0)^{1/2})^2} \leq \phi''(t) \leq \frac{\phi''(0)}{(1 - t\phi''(0)^{1/2})^2} \tag{23}$$

where the lower and the uppers bounds hold for $t \geq 0$ and $t \in [0, \phi''(0)^{-1/2})$, with $t \in \text{dom } \phi$, respectively (see also Boyd & Vandenberghe (2004)). Consider now self-concordant functions on $\mathbb{R}^n$ and let $S(\mathbf{x}) = \{\mathbf{y} \in \mathbb{R}^n : \|\mathbf{y} - \mathbf{x}\|_{\mathbf{x}} < 1\}$. For any $\mathbf{x} \in \text{dom } f$ and $\mathbf{y} \in S(\mathbf{x})$, we have that (Nesterov (2004))

$$(1 - \|\mathbf{y} - \mathbf{x}\|_{\mathbf{x}})^2 \nabla^2 f(\mathbf{x}) \preceq \nabla^2 f(\mathbf{y}) \preceq \frac{1}{(1 - \|\mathbf{y} - \mathbf{x}\|_{\mathbf{x}})^2} \nabla^2 f(\mathbf{x}). \tag{24}$$

The following results will be used later for the convergence analysis of the algorithms. For their proofs see Nesterov (2004).

**Lemma A.1** ((Nesterov, 2004)). *Let* $\mathbf{x}, \mathbf{y} \in \text{dom } f$. *If* $\|\mathbf{y} - \mathbf{x}\|_x < 1$, *then,*

$$f(\mathbf{y}) \leq f(\mathbf{x}) + \langle \nabla f(\mathbf{x}), \mathbf{y} - \mathbf{x} \rangle + \omega_*(\|\mathbf{y} - \mathbf{x}\|_{\mathbf{x}})$$

**Lemma A.2** (Nesterov (2004)). *Let* $\mathbf{x}, \mathbf{y} \in \text{dom } f$. *Then,*

$$f(\mathbf{y}) \geq f(\mathbf{x}) + \langle \nabla f(\mathbf{x}), \mathbf{y} - \mathbf{x} \rangle + \omega(\|\nabla f(\mathbf{y}) - \nabla f(\mathbf{x})\|_{\mathbf{y}}^*).$$

*If in addition* $\|\nabla f(\mathbf{y}) - \nabla f(\mathbf{x})\|_{\mathbf{y}}^* < 1$, *then,*

$$f(\mathbf{y}) \leq f(\mathbf{x}) + \langle \nabla f(\mathbf{x}), \mathbf{y} - \mathbf{x} \rangle + \omega_*(\|\nabla f(\mathbf{y}) - \nabla f(\mathbf{x})\|_{\mathbf{y}}^*).$$

Throughout this paper, we refer to notions such as super-linear and quadratic convergence rates. Denote $\mathbf{x}_k$ the iterate generated by an iterative process at the $k^{\text{th}}$ iteration. The sub-optimality gap of the Newton method for self-concordant function satisfies the bound $f(\mathbf{x}_k) - f(\mathbf{x}^*) \leq \lambda_f(\mathbf{x}_k)^2$ which holds for $\lambda_f(\mathbf{x}_k) \leq 0.68$ (Boyd & Vandenberghe (2004)) and thus one can estimate the convergence rate in terms of the local norm of the gradient. It is known that the Newton method achieves a local quadratic convergence rate. In this setting, we say that a process converges quadratically if $\lambda_f(\mathbf{x}_{k+1}) \leq R\lambda_f(\mathbf{x}_k)^2$, for $R > 0$. In addition to quadratic converge rate, we say that a process converges with super-linear rate if $\lambda_f(\mathbf{x}_{k+1})/\lambda_f(\mathbf{x}_k) \leq R(k)$, where $R(k) \downarrow 0$.

## B   Proof of Theorem 3.1

The update rule of the Low-Rank Newton method at iteration $k$ is

$$\mathbf{x}_{k+1} = \mathbf{x}_k + t_k \hat{\mathbf{d}}_{h,k}, \tag{25}$$

where for the purposes of this section we use $\hat{\mathbf{d}}_{h,k} := -\mathbf{Q}_{h,k}^{-1} \nabla f(\mathbf{x}_k)$, where $\mathbf{Q}_{h,k}^{-1}$ is defined in equation 10. Define also the corresponding approximate Newton decrement $\bar{\lambda}(\mathbf{x}) := (\nabla f(\mathbf{x}_k)^T \mathbf{Q}_{h,k}^{-1} \nabla f(\mathbf{x}_k))^{\frac{1}{2}}$.

**Lemma B.1.** *For any $k \in \mathbb{N}$ it holds that*

$$\bar{\lambda}(\mathbf{x}_k)^2 = -\nabla f(\mathbf{x}_k)^T \hat{\mathbf{d}}_{h,k} \tag{26}$$

$$\sqrt{\frac{\sigma_{k,n}}{\sigma_{k,N+1}}} \lambda(\mathbf{x}_k) \leq \bar{\lambda}(\mathbf{x}_k) \leq \lambda(\mathbf{x}_k) \tag{27}$$

$$\|\hat{\mathbf{d}}_{h,k}\|_{\mathbf{x}} \leq \bar{\lambda}(\mathbf{x}_k) \tag{28}$$

$$\left\| [\nabla^2 f(\mathbf{x}_k)]^{1/2} \left( \hat{\mathbf{d}}_{h,k} - \mathbf{d}_{h,k} \right) \right\|_2 \leq \left( 1 - \frac{\sigma_{k,n}}{\sigma_{k,N+1}} \right) \lambda(\mathbf{x}_k). \tag{29}$$

*where $\| \cdot \|_{\mathbf{x}}$ is defined in equation 20.*

*Proof.* Equality equation 26 follows directly by the definition of $\hat{\mathbf{d}}_{h,k}$. For the right-hand side of equation 27 we get

$$\bar{\lambda}(\mathbf{x}_k)^2 - \lambda(\mathbf{x}_k)^2 = \nabla f(\mathbf{x}_k)^T (\mathbf{Q}_{h,k}^{-1} - \nabla f^2(\mathbf{x}_k)^{-1}) \nabla f(\mathbf{x}_k)^T$$
$$= \nabla f(\mathbf{x}_k)^T [\nabla f^2(\mathbf{x}_k)]^{-\frac{1}{2}} ([\nabla f^2(\mathbf{x}_k)]^{\frac{1}{2}} \mathbf{Q}_{h,k}^{-1} [\nabla f^2(\mathbf{x}_k)]^{\frac{1}{2}} - \mathbf{I}_n)[\nabla f^2(\mathbf{x}_k)]^{-\frac{1}{2}} \nabla f(\mathbf{x}_k).$$

By construction of $\mathbf{Q}_{h,k}^{-1}$, it holds that

$$[\nabla f^2(\mathbf{x}_k)]^{\frac{1}{2}} \mathbf{Q}_{h,k}^{-1} [\nabla f^2(\mathbf{x}_k)]^{\frac{1}{2}} - \mathbf{I}_n \preceq \sigma_{\max}([\nabla f^2(\mathbf{x}_k)]^{\frac{1}{2}} \mathbf{Q}_{h,k}^{-1} [\nabla f^2(\mathbf{x}_k)]^{\frac{1}{2}} - \mathbf{I}_n)\mathbf{I}_n$$
$$= \max_{i=1,\dots n} \left\{ \frac{\sigma_i(\nabla f^2(\mathbf{x}_k))}{\sigma_i(\mathbf{Q}_{h,k})} - 1 \right\} = 0,$$

and thus $\bar{\lambda}(\mathbf{x}_k) - \lambda(\mathbf{x}_k) \leq 0$. Using similar arguments we take

$$\lambda(\mathbf{x}_k)^2 - \bar{\lambda}(\mathbf{x}_k)^2 \leq \max_{i=1,\dots n} \left\{ 1 - \frac{\sigma_i(\nabla f^2(\mathbf{x}_k))}{\sigma_i(\mathbf{Q}_{h,k})} \right\} \lambda(\mathbf{x})^2$$
$$= \left( 1 - \frac{\sigma_{k,n}}{\sigma_{k,N+1}} \right) \lambda(\mathbf{x})^2$$

and thus the left-hand side inequality of equation 27 follows directly. As for the result in equation 28, using the fact that $\mathbf{Q}_{h,k}$ is symmetric positive-definite, we take

$$\|\hat{\mathbf{d}}_{h,k}\|_{\mathbf{x}}^2 = \hat{\mathbf{d}}_{h,k}^T \nabla f^2(\mathbf{x}_k) \hat{\mathbf{d}}_{h,k}$$
$$= \nabla f(\mathbf{x}_k)^T [\mathbf{Q}_{h,k}]^{-\frac{1}{2}} \left( [\mathbf{Q}_{h,k}]^{-\frac{1}{2}} \nabla f^2(\mathbf{x}_k) [\mathbf{Q}_{h,k}]^{-\frac{1}{2}} \right) [\mathbf{Q}_{h,k}]^{-\frac{1}{2}} f(\mathbf{x}_k)$$
$$\leq \sigma_{\max} \left( [\mathbf{Q}_{h,k}]^{-\frac{1}{2}} \nabla f^2(\mathbf{x}_k) [\mathbf{Q}_{h,k}]^{-\frac{1}{2}} \right) \bar{\lambda}(\mathbf{x}_k)^2,$$

and since $\sigma_{\max}\left([\mathbf{Q}_{h,k}]^{-\frac{1}{2}}\nabla f^2(\mathbf{x}_k)[\mathbf{Q}_{h,k}]^{-\frac{1}{2}}\right) = 1$ the result follows. Last, we prove equation 29. Using the definitions of $\hat{\mathbf{d}}_{h,k}$ and $\hat{d}_{h,k}$ we have

$$
\begin{aligned}
\left\|[\nabla^2 f(\mathbf{x}_k)]^{1/2}\left(\hat{\mathbf{d}}_{h,k} - \mathbf{d}_{h,k}\right)\right\|_2 &= \left\|[\nabla^2 f(\mathbf{x}_k)]^{\frac{1}{2}}([\nabla^2 f(\mathbf{x}_k)]^{-1}) - [\mathbf{Q}_{h,k}]^{-1})\nabla f(\mathbf{x}_k)\right\|_2 \\
&\leq \left\|\mathbf{I}_n - [\nabla^2 f(\mathbf{x}_k)]^{\frac{1}{2}}[\mathbf{Q}_{h,k}]^{-1}[\nabla^2 f(\mathbf{x}_k)]^{\frac{1}{2}}\right\|_2 \bar{\lambda}(\mathbf{x}) \\
&= \max\left\{1 - \frac{\sigma_1}{\sigma_1}, \ldots, 1 - \frac{\sigma_{N+1}}{\sigma_{N+1}}, \ldots, 1 - \frac{\sigma_n}{\sigma_{N+1}}\right\} \bar{\lambda}(\mathbf{x}) \\
&= \left(1 - \frac{\sigma_n}{\sigma_{N+1}}\right)\bar{\lambda}(\mathbf{x})
\end{aligned}
$$

which concludes the proof of the lemma. $\qquad\square$

**Lemma B.2.** *Let $\lambda(\mathbf{x}_k) > \eta$ for some $\eta \in (0, \frac{3-\sqrt{5}}{2})$. Then, there exists $\gamma > 0$ such that the coarse direction $\hat{\mathbf{d}}_{h,k}$ will yield reduction in value function*

$$
f(\mathbf{x}_k + t_k\hat{\mathbf{d}}_{h,k}) - f(\mathbf{x}_k) \leq -\gamma,
$$

*for any $k \in \mathbb{N}$.*

*Proof.* By Lemma A.1 together with equation 25 and equation 26 we have that

$$
\begin{aligned}
f(\mathbf{x}_k + t_k\hat{\mathbf{d}}_{h,k}) &\leq f(\mathbf{x}_k) + \langle\nabla f(\mathbf{x}_k), \mathbf{x}_{k+1} - \mathbf{x}_k\rangle + \omega_*(\|\mathbf{x}_{k+1} - \mathbf{x}_k\|_{\mathbf{x}_k}) \\
&= f(\mathbf{x}_k) + t_k\nabla f(\mathbf{x}_k)^T\hat{\mathbf{d}}_{h,k} + \omega_*(t_k\|\hat{\mathbf{d}}_{h,k}\|_{\mathbf{x}_k}) \\
&\leq f(\mathbf{x}_k) - t_k\bar{\lambda}(\mathbf{x})^2 + \omega_*(t_k\bar{\lambda}(\mathbf{x}))
\end{aligned}
$$

where the last inequality follows from equation 28 and the fact that $\omega_*(x)$ is a monotone increasing function. Note that the above inequality is valid for $t_{h,k} < \frac{1}{\bar{\lambda}(\mathbf{x}_{h,k})}$. Next, the right-hand side is minimized at $t_h^* = \frac{1}{1+\bar{\lambda}(\mathbf{x}_{h,k})}$ and thus

$$
\begin{aligned}
f(\mathbf{x}_k + t^*\hat{\mathbf{d}}_{h,k}) &\leq f(\mathbf{x}_k) - \frac{\bar{\lambda}(\mathbf{x})^2}{1+\bar{\lambda}(\mathbf{x}_{h,k})} + \omega_*\left(\frac{\bar{\lambda}(\mathbf{x})}{1+\bar{\lambda}(\mathbf{x}_{h,k})}\right) \\
&= f(\mathbf{x}_k) - \bar{\lambda}(\mathbf{x}_k) + \log\left(1 + \bar{\lambda}(\mathbf{x}_k)\right).
\end{aligned}
$$

Using the inequality $-x + \log(1+x) \leq -\frac{x^2}{2(1+x)}$ for any $x > 0$, we obtain

$$
\begin{aligned}
f(\mathbf{x}_k + t^*\hat{\mathbf{d}}_{h,k}) &\leq f(\mathbf{x}_k) - \frac{\bar{\lambda}(\mathbf{x}_k)^2}{2(1+\bar{\lambda}(\mathbf{x}_k))} \\
&\leq f(\mathbf{x}_k) - \alpha t^*\bar{\lambda}(\mathbf{x}_k)^2 \\
&= f(\mathbf{x}_k) + \alpha t^*\nabla f(\mathbf{x}_k)^T\hat{\mathbf{d}}_{h,k},
\end{aligned}
$$

thus $t^*$ satisfies the back-tracking line search exit condition and that it will always return a step size $t_k > \beta/(1+\hat{\lambda}(\mathbf{x}_k))$. Therefore,

$$
f_h(\mathbf{x}_k + t_k\hat{\mathbf{d}}_{h,k}) - f(\mathbf{x}_k) \leq -\alpha\beta\frac{\bar{\lambda}(\mathbf{x}_k)^2}{1+\bar{\lambda}(\mathbf{x}_k)}.
$$

Additionally, combining the left-hand side in equation 27, the fact that the function $x \to \frac{x^2}{1+x}$ is monotone increasing for any $x > 0$ and since by assumption $\lambda(\mathbf{x}_k) > \eta$ we obtain

$$
f(\mathbf{x}_k + t_k\hat{\mathbf{d}}_{h,k}) - f(\mathbf{x}_k) \leq -\alpha\beta\frac{\eta^2}{1+\eta}.
$$

which concludes the proof by setting $\gamma := \alpha\beta\eta^2/(1+\eta)$. $\qquad\square$

Next, we estimate the sub-optimality gap of the process.

**Lemma B.3.** *Let $\lambda(\mathbf{x}_k) < 1$. Then,*

$$\omega(\lambda(\mathbf{x}_k)) \leq f(\mathbf{x}_k) - f(\mathbf{x}^*) \leq \omega_*(\lambda(\mathbf{x}_k)).$$

*If in addition $\lambda(\mathbf{x}_k) < 0.68$ then,*

$$f(\mathbf{x}_k) - f(\mathbf{x}^*) \leq \lambda(\mathbf{x}_k)^2.$$

*Proof.* The first result follows directly from Lemma A.2 Further, it holds that $\omega_*(x) \leq x^2, 0 \leq x \leq 0.68$ and thus we take the second result. $\qquad\square$

Therefore, $\lambda(\mathbf{x}_k)^2 \leq \epsilon, \epsilon \in (0, 0.68^2)$, can be used as an exit condition for the Low-Rank Newton method. Next, we prove that the line search selects the unit step.

**Lemma B.4.** *If $\bar{\lambda}(\mathbf{x}_k) \leq (1 - 2\alpha)/2$ then the Low-Rank Newton method accepts the unit step, $t_k = 1$.*

*Proof.* By Lemma A.1 we have

$$f(\mathbf{x}_k + \hat{\mathbf{d}}_{h,k}) \leq f(\mathbf{x}_k) - \bar{\lambda}(\mathbf{x})^2 + \omega_*(\bar{\lambda}(\mathbf{x}))$$
$$= f(\mathbf{x}_k) - \bar{\lambda}(\mathbf{x})^2 - \bar{\lambda}(\mathbf{x}) - \log(1 - \bar{\lambda}(\mathbf{x}))$$

which holds since by assumption $\bar{\lambda}(\mathbf{x}) < 1$. Further, $-x - \log(1 - x)$ for all $x \in (0, 0.81)$ and hence

$$f(\mathbf{x}_k + \hat{\mathbf{d}}_{h,k}) \leq f(\mathbf{x}_k) - \bar{\lambda}(\mathbf{x})^2 + \frac{1}{2}\bar{\lambda}(\mathbf{x})^2 + \bar{\lambda}(\mathbf{x})^3$$
$$= f(\mathbf{x}_k) - \frac{1}{2}(1 - 2\bar{\lambda}(\mathbf{x}))\bar{\lambda}(\mathbf{x})^2.$$

Therefore, if $\bar{\lambda}(\mathbf{x}_k) \leq (1 - 2\alpha)/2$ we take

$$f(\mathbf{x}_k + \hat{\mathbf{d}}_{h,k}) \leq f(\mathbf{x}_k) - \alpha\bar{\lambda}(\mathbf{x})^2$$

which satisfies the line search condition for $t_k = 1$. $\qquad\square$

**Lemma B.5.** *Let $f : \mathbb{R}^n \to \mathbb{R}$ be a strictly convex self-concordant function. If $\bar{\lambda}(\mathbf{x}_k) < \frac{1}{t_k}$, we have that*

*(i)* $\nabla^2 f(\mathbf{x}_{k+1}) \preceq \frac{1}{(1 - t_k\bar{\lambda}(\mathbf{x}_k))^2}\nabla^2 f(\mathbf{x}_k)$,

*(ii)* $[\nabla^2 f(\mathbf{x}_{k+1})]^{-1} \preceq \frac{1}{(1 - t_k\bar{\lambda}(\mathbf{x}_k))^2}[\nabla^2 f(\mathbf{x}_k)]^{-1}$.

*Proof.* The proof is analogue to Lemma C.3 below but with $\bar{\lambda}(\mathbf{x}_k)$ instead of $\hat{\lambda}(\mathbf{x}_k)$. $\qquad\square$

The next lemma shows the local super-linear rate of the Low-Rank Newton method.

**Lemma B.6.** *Suppose that the sequence is generated by equation 25 with $t_k = 1$. Then,*

$$\lambda(\mathbf{x}_{k+1}) \leq \frac{1 - \frac{\sigma_{k,n}}{\sigma_{k,N+1}} + \lambda(\mathbf{x}_k)}{(1 - \lambda(\mathbf{x}_k))^2}\lambda(\mathbf{x}_k).$$

*Proof.* By Lemma B.5, the upper bound in equation 27 and $t_k = 1$ we have

$$\nabla^2 f(\mathbf{x}_{k+1}) \preceq \frac{1}{(1 - \lambda(\mathbf{x}_k))^2}\nabla^2 f(\mathbf{x}_k). \tag{30}$$

Now, by the definition of the Newton decrement we have that

$$\lambda(\mathbf{x}_{k+1}) = \left[\nabla f(\mathbf{x}_{k+1})^T[\nabla^2 f(\mathbf{x}_{k+1})]^{-1}\nabla f(\mathbf{x}_{k+1})\right]^{1/2},$$

and combining this with inequality equation 30 we take

$$\lambda(\mathbf{x}_{k+1}) \leq \frac{1}{1 - \lambda(\mathbf{x}_k)} \left\| [\nabla^2 f(\mathbf{x}_k)]^{-1/2} \nabla f(\mathbf{x}_{k+1}) \right\|_2. \tag{31}$$

Denote $\mathbf{Z} := [\nabla^2 f(\mathbf{x}_k)]^{-1/2} \nabla f(\mathbf{x}_{k+1})$. Using the fact that

$$\nabla f(\mathbf{x}_{k+1}) = \int_0^1 \nabla^2 f(\mathbf{x}_k + y\hat{\mathbf{d}}_{h,k})\hat{\mathbf{d}}_{h,k} \, dy + \nabla f(\mathbf{x}_k)$$

we see that

$$\mathbf{Z} = [\nabla^2 f(\mathbf{x}_k)]^{-1/2} \left( \int_0^1 \nabla^2 f(\mathbf{x}_k + y\hat{\mathbf{d}}_{h,k})\hat{\mathbf{d}}_{h,k} \, dy + \nabla f(\mathbf{x}_k) \right)$$

$$= \underbrace{\int_0^1 [\nabla^2 f(\mathbf{x}_k)]^{-1/2} \nabla^2 f(\mathbf{x}_k + y\hat{\mathbf{d}}_{h,k})[\nabla^2 f(\mathbf{x}_k)]^{-1/2} \, dy}_{\mathbf{T}} \, [\nabla^2 f(\mathbf{x}_k)]^{1/2}\hat{\mathbf{d}}_{h,k} - [\nabla^2 f(\mathbf{x}_k)]^{1/2}\mathbf{d}_{h,k},$$

where $\mathbf{d}_{h,k}$ is the Newton direction. Next adding and subtracting the quantity $\mathbf{T}[\nabla^2 f(\mathbf{x}_k)]^{1/2}\mathbf{d}_{h,k}$ we have that

$$\mathbf{Z} = \mathbf{T}[\nabla^2 f(\mathbf{x}_k)]^{1/2}(\hat{\mathbf{d}}_{h,k} - \mathbf{d}_{h,k}) + (\mathbf{T} - \mathbf{I}_{N \times N})[\nabla^2 f(\mathbf{x}_k)]^{1/2}\mathbf{d}_{h,k},$$

and thus

$$\|\mathbf{Z}\| \leq \left\| \underbrace{\mathbf{T}[\nabla^2 f(\mathbf{x}_k)]^{1/2}(\hat{\mathbf{d}}_{h,k} - \mathbf{d}_{h,k})}_{\mathbf{Z}_1} \right\|_2 + \left\| \underbrace{(\mathbf{T} - \mathbf{I}_{N \times N})[\nabla^2 f(\mathbf{x}_k)]^{1/2}\mathbf{d}_{h,k}}_{\mathbf{Z}_2} \right\|_2 \tag{32}$$

Using again equation 30 and since, by assumption, $y\hat{\lambda}(\mathbf{x}_k) < 1$ we take

$$[\nabla^2 f(\mathbf{x}_k)]^{-1/2} \nabla^2 f(\mathbf{x}_k + y\hat{\mathbf{d}}_{h,k})[\nabla^2 f(\mathbf{x}_k)]^{-1/2} \preceq \frac{1}{(1 - y\lambda(\mathbf{x}_k))^2}\mathbf{I}_{N \times N}.$$

We are now in position to estimate both norms in equation 32. For the first one we have that

$$\|\mathbf{Z}_1\| \leq \left\| \int_0^1 \frac{1}{(1 - y\lambda(\mathbf{x}_k))^2} dy \right\|_2 \left\| [\nabla^2 f(\mathbf{x}_k)]^{1/2} \left( \hat{\mathbf{d}}_{h,k} - \mathbf{d}_{h,k} \right) \right\|_2$$

$$= \frac{1}{1 - \lambda(\mathbf{x}_k)} \left\| [\nabla^2 f(\mathbf{x}_k)]^{1/2} \left( \hat{\mathbf{d}}_{h,k} - \mathbf{d}_{h,k} \right) \right\|_2$$

$$\leq \left( 1 - \frac{\sigma_{k,n}}{\sigma_{k,N+1}} \right) \frac{\lambda(\mathbf{x}_k)}{1 - \lambda(\mathbf{x}_{h,k})}.$$

where the last inequality follows from equation 29. Next, the second norm implies

$$\|\mathbf{Z}_2\| \leq \left\| \int_0^1 \left( \frac{1}{(1 - y\lambda(\mathbf{x}_k))^2} - 1 \right) dy \right\|_2 \left\| [\nabla^2 f(\mathbf{x}_k)]^{1/2}\mathbf{d}_{h,k} \right\|_2$$

$$= \frac{\lambda(\mathbf{x}_k)}{1 - \lambda(\mathbf{x}_k)}\lambda(\mathbf{x}_k).$$

Putting this all together, inequality equation 31 becomes

$$\lambda(\mathbf{x}_{k+1}) \leq \frac{1 - \frac{\sigma_{k,n}}{\sigma_{k,N+1}} + \lambda(\mathbf{x}_k)}{(1 - \lambda(\mathbf{x}_k))^2}\lambda(\mathbf{x}_k).$$

as required. $\qquad\qquad\qquad\qquad\qquad\qquad\qquad\qquad\qquad\qquad\qquad\qquad\qquad\qquad\qquad\qquad\qquad$ □

Next we prove theorem 3.1. Recall the

*Proof of theorem 3.1.* By lemma B.2, we see that one step of the first phase of the Low-Rank Newton method decreases the objective by $\gamma := \alpha\beta\eta^2/(1+\eta) > 0$. By Lemma B.2 we also note that the line-search can be applied to determine the step size parameter. This proves the result of phase (i) of the Low-Rank Newton method.

Next, in phase (ii) we have $\lambda(\mathbf{x}_k) \leq \eta$. Further, by lemma B.4, and since $\lambda(\mathbf{x}_k) \leq \eta$, the line search accepts the unit step. Moreover, by strict convexity, $\varepsilon_k > 0$ for all $k \in \mathbb{N}$. If $\varepsilon \in (0,1)$, then $\varepsilon_{\min} \in (0,1]$ Hence, by Lemma B.6, since $\frac{\sigma_{k,n}}{\sigma_{k,N+1}} \geq \varepsilon_{\min}$ and $\lambda(\mathbf{x}_k) \leq \eta$, we take

$$\lambda(\mathbf{x}_{k+1}) \leq \frac{1 - \varepsilon_{\min} + \eta}{(1-\eta)^2} \lambda(\mathbf{x}_k).$$

Moreover, by definition $\eta := (3 - \sqrt{9-4\varepsilon})$ and since $0 < \varepsilon_{\min} \leq 1$ we obtain $\frac{1-\varepsilon_{\min}+\hat{\eta}}{(1-\hat{\eta})^2} < 1$. This shows that $\lambda(\mathbf{x}_{k+1}) < \lambda(\mathbf{x}_k)$ and, in particular, this process converges with composite rate for $\lambda(\mathbf{x}_k) \leq \eta$ and some $\eta \in (0, \frac{3-\sqrt{5}}{2})$.

Let $\varepsilon = 1$. Then, $\lim_{k\to\infty} \varepsilon_k = 1$ since $(\varepsilon_k)_{k\in\mathbb{N}}$ is bounded and $\varepsilon \leq \limsup_{k\to\infty} \varepsilon_k \leq 1$. Using again the result of lemma B.6 together with $\lambda(\mathbf{x}_k) \leq \eta$ we have that $\frac{1-\varepsilon_k+\hat{\eta}}{(1-\hat{\eta})^2} < 1$, for all $k \in \mathbb{N}$. Therefore, $\lambda(\mathbf{x}_{k+1}) < \lambda(\mathbf{x}_k)$. This means

$$\lim_{k\to\infty} \frac{1 - \varepsilon_k + \lambda(\mathbf{x}_k)}{(1-\lambda(\mathbf{x}_k))^2} = 0,$$

which proves the super-linear rate for the low-rank Newton method as required. $\qquad\square$

## C  Proof of Theorem 3.2

Let $f$ be a strictly convex self-concordant function. Theorem 3.2 considers the sequence generated by

$$\mathbf{x}_{k+1} = \mathbf{x}_k - t_k \mathbf{P}_k \mathbf{Q}_{H,k}^{-1} \mathbf{R}_k \nabla f(\mathbf{x}_k).$$

For the purposes of this section we denote $\hat{\mathbf{d}}_{h,k} := -\mathbf{P}_k \mathbf{Q}_{H,k}^{-1} \mathbf{R}_k \nabla f(\mathbf{x}_k)$, where $\mathbf{Q}_{H,k}^{-1}$ is defined in equation 13 and $\mathbf{P}_k$ in definition 1. Below we will make use of the approximate decrements $\hat{\lambda}(\mathbf{x})$ and $\tilde{\lambda}(\mathbf{x})$ defined in equation 14, respectively. Before proving Lemma 3.1 we state an upper bound for $\hat{\lambda}(\mathbf{x})$ which was proved in Tsipinakis & Parpas (2024).

**Lemma C.1** ((Tsipinakis & Parpas, 2024)). *Let $\lambda(\mathbf{x}_{h,k})$ be the newton decrement in equation 21. For any $k \in \mathbb{N}$ it holds that*

$$\tilde{\lambda}(\mathbf{x}_{h,k}) \leq \lambda(\mathbf{x}_{h,k}). \tag{33}$$

Note that the above result holds with probability one. Below we prove Lemma 3.1. We state again Lemma 3.1 for convenience.

**Lemma C.2.** *For all $k \in \mathbb{N}$ we have*

$$\mathbf{d}_{h,k}^T \nabla^2 f(\mathbf{x}_k) \hat{\mathbf{d}}_{h,k} = \hat{\lambda}^2(\mathbf{x}_k) \tag{34}$$

$$\hat{\mathbf{d}}_{h,k}^T \nabla^2 f(\mathbf{x}_k) \hat{\mathbf{d}}_{h,k} \leq \hat{\lambda}^2(\mathbf{x}_k) \tag{35}$$

$$\|[\nabla^2 f(\mathbf{x}_k)]^{\frac{1}{2}} (\mathbf{d}_{h,k} - \hat{\mathbf{d}}_{h,k})\| \leq \hat{e}_k \tag{36}$$

$$\sqrt{\frac{\sigma_{k,N}}{\sigma_{k,p+1}}} \tilde{\lambda}(\mathbf{x}_k) \leq \hat{\lambda}(\mathbf{x}_k) \leq \tilde{\lambda}(\mathbf{x}_k) \leq \lambda(\mathbf{x}_k), \tag{37}$$

*where $\hat{e}_k := \sqrt{\lambda(\mathbf{x}_k)^2 - \hat{\lambda}(\mathbf{x}_k)^2}$.*

*Further, let Assumption 2 hold. Then, there exists $\mu_k \in (0,1]$ such that $\hat{e}_k \leq (1 - \frac{\sigma_{k,N}}{\sigma_{k,p+1}} \mu_k^2)^{1/2} \lambda(\mathbf{x}_k)$ with probability $\delta$.*

*Proof of Lemma 3.1.* Equality equation 34 follows directly from the definitions of $\hat{\mathbf{d}}_{h,k}$ and $\mathbf{d}_{h,k}$. By equation 34 we also get $\hat{\lambda}^2(\mathbf{x}_k) = -\nabla f(\mathbf{x}_k)^T \hat{\mathbf{d}}_{h,k}$. For inequality equation 35, since $\mathbf{Q}_{H,k}$ is symmetric positive definite, we have

$$\hat{\mathbf{d}}_{h,k}^T \nabla^2 f(\mathbf{x}_k) \hat{\mathbf{d}}_{h,k} = \nabla f(\mathbf{x}_k)^T \mathbf{P}_k \mathbf{Q}_{H,k}^{-1} \mathbf{R}_k \nabla^2 f(\mathbf{x}_k) \mathbf{P}_k \mathbf{Q}_{H,k}^{-1} \mathbf{R}_k \nabla f(\mathbf{x}_k)$$
$$= \mathbf{z}^T \mathbf{Q}_{H,k}^{-\frac{1}{2}} (\mathbf{R}_k \nabla^2 f(\mathbf{x}_k) \mathbf{P}_k) \mathbf{Q}_{H,k}^{-\frac{1}{2}} \mathbf{z}$$

where $\mathbf{z} := \mathbf{Q}_{H,k}^{-\frac{1}{2}} \mathbf{R}_k \nabla f(\mathbf{x}_k)$ and note that $\mathbf{z}^T \mathbf{z} = \hat{\lambda}^2(\mathbf{x}_k)$. By construction of $\mathbf{Q}_{H,k}^{-1}$, it holds that

$$\mathbf{Q}_{H,k}^{-\frac{1}{2}} (\mathbf{R}_k \nabla^2 f(\mathbf{x}_k) \mathbf{P}_k) \mathbf{Q}_{H,k}^{-\frac{1}{2}} \preceq \sigma_{\max}(\mathbf{Q}_{H,k}^{-\frac{1}{2}} (\mathbf{R}_k \nabla^2 f(\mathbf{x}_k) \mathbf{P}_k) \mathbf{Q}_{H,k}^{-\frac{1}{2}}) \mathbf{I}_N$$
$$= \max \left\{ \frac{\sigma_{k,1}}{\sigma_{k,1}}, \dots, \frac{\sigma_{k,p+1}}{\sigma_{k,p+1}}, \frac{\sigma_{k,p+2}}{\sigma_{k,p+1}}, \dots, \frac{\sigma_{k,N}}{\sigma_{k,p+1}} \right\} \mathbf{I}_N$$
$$= \mathbf{I}_N$$

Then, $\mathbf{z}^T \mathbf{Q}_{H,k}^{-\frac{1}{2}} (\mathbf{R}_k \nabla^2 f(\mathbf{x}_k) \mathbf{P}_k) \mathbf{Q}_{H,k}^{-\frac{1}{2}} \mathbf{z} \leq \hat{\lambda}^2(\mathbf{x}_k)$ and equation 35 is proved. Next,

$$\|[\nabla^2 f(\mathbf{x}_k)]^{\frac{1}{2}} (\mathbf{d}_{h,k} - \hat{\mathbf{d}}_{h,k})\|^2 = \mathbf{d}_{h,k}^T \nabla^2 f(\mathbf{x}_k) \mathbf{d}_{h,k} - 2 \mathbf{d}_{h,k}^T \nabla^2 f(\mathbf{x}_k) \hat{\mathbf{d}}_{h,k} + \hat{\mathbf{d}}_{h,k}^T \nabla^2 f(\mathbf{x}_k) \hat{\mathbf{d}}_{h,k}$$
$$\leq \lambda(\mathbf{x}_k)^2 - \hat{\lambda}(\mathbf{x}_k)^2,$$

where the last inequality follows from equation 21, equation 34 and equation 35. This proves inequality equation 36. For the bounds in equation 37 we start by proving $\hat{\lambda}(\mathbf{x}_k) \leq \tilde{\lambda}(\mathbf{x}_k)$. We have

$$\hat{\lambda}(\mathbf{x}_k)^2 - \tilde{\lambda}(\mathbf{x}_k)^2 = (\mathbf{R}\nabla f(\mathbf{x}_k))^T (\mathbf{Q}_{H,k}^{-1} - [\mathbf{R}_k \nabla^2 f(\mathbf{x}_k) \mathbf{P}_k]^{-1})(\mathbf{R}_k \nabla f(\mathbf{x}_k))$$
$$= (\mathbf{Q}_{H,k}^{-\frac{1}{2}} \mathbf{R}_k \nabla f(\mathbf{x}_k))^T (\mathbf{I}_N - \mathbf{Q}_{H,k}^{\frac{1}{2}} [\mathbf{R}_k \nabla^2 f(\mathbf{x}_k) \mathbf{P}_k]^{-1} \mathbf{Q}_{H,k}^{\frac{1}{2}})(\mathbf{Q}_{H,k}^{-\frac{1}{2}} \mathbf{R}\nabla f(\mathbf{x}_k))$$

We also have

$$\mathbf{I}_N - \mathbf{Q}_{H,k}^{\frac{1}{2}} [\mathbf{R}\nabla^2 f(\mathbf{x}_k) \mathbf{P}_k]^{-1} \mathbf{Q}_{H,k}^{\frac{1}{2}} \preceq \sigma_{\max}(\mathbf{I}_N - \mathbf{Q}_{H,k}^{\frac{1}{2}} [\mathbf{R}_k \nabla^2 f(\mathbf{x}_k) \mathbf{P}_k]^{-1} \mathbf{Q}_{H,k}^{\frac{1}{2}}) \mathbf{I}_N$$
$$= \max \left\{ 1 - \frac{\sigma_{k,1}}{\sigma_{k,1}}, \dots, 1 - \frac{\sigma_{k,p+1}}{\sigma_{k,p+1}}, \dots, 1 - \frac{\sigma_{k,p+1}}{\sigma_{k,N}} \dots \right\} \mathbf{I}_N$$
$$= 0.$$

Putting this all together, we take $\hat{\lambda}(\mathbf{x}_k) \leq \tilde{\lambda}(\mathbf{x}_k)$. On the other hand, with similar arguments we have that

$$\tilde{\lambda}(\mathbf{x}_k)^2 - \hat{\lambda}(\mathbf{x}_k)^2 = \mathbf{z}^T (\mathbf{Q}_{H,k}^{\frac{1}{2}} [\mathbf{R}_k \nabla^2 f(\mathbf{x}_k) \mathbf{P}_k]^{-1} \mathbf{Q}_{H,k}^{\frac{1}{2}} - \mathbf{I}_N) \mathbf{z}$$
$$\leq \left( \frac{\sigma_{k,p+1}}{\sigma_{k,N}} - 1 \right) \mathbf{z}^T \mathbf{z} = \left( \frac{\sigma_{k,p+1}}{\sigma_{k,N}} - 1 \right) \hat{\lambda}(\mathbf{x}_k)^2$$

and thus the lower bound of equation 37 follows. Putting this all together and combining it with Lemma C.1, equation 37 has been proved. Moreover, assumption 2 and equation 37 imply $\tilde{\lambda}(\mathbf{x}_k) > 0$. Then there exists $\mu_k \in (0, \frac{\tilde{\lambda}(\mathbf{x}_k)}{\lambda(\mathbf{x}_k)}]$ with probability $\delta$. Thus, we obtain

$$\mu_k \sqrt{\frac{\sigma_{k,N}}{\sigma_{k,p+1}}} \lambda(\mathbf{x}_k) \leq \hat{\lambda}(\mathbf{x}_k) \leq \lambda(\mathbf{x}_k), \tag{38}$$

with probability $\delta$. Therefore, by the lower bound of the last result we directly get $\hat{e}_k \leq (1 - \frac{\sigma_{k,N}}{\sigma_{k,p+1}} \mu_k^2)^{1/2} \lambda(\mathbf{x}_k)$, with probability $\delta$ too, which concludes the proof. $\square$

**Lemma C.3.** *Let* $f : \mathbb{R}^n \to \mathbb{R}$ *be a strictly convex self-concordant function and* $\mathbf{x}_{k+1} = \mathbf{x}_k - t_k \mathbf{P}_k \mathbf{Q}_{H,k}^{-1} \mathbf{R}_k \nabla f(\mathbf{x}_k)$. *If* $\hat{\lambda}(\mathbf{x}_{h,k}) < \frac{1}{t_{h,k}}$, *we have that*

*(i)* $\nabla^2 f(\mathbf{x}_{k+1}) \preceq \frac{1}{(1-t_k\hat{\lambda}(\mathbf{x}_k))^2}\nabla^2 f(\mathbf{x}_k)$,

*(ii)* $[\nabla^2 f(\mathbf{x}_{k+1})]^{-1} \preceq \frac{1}{(1-t_k\hat{\lambda}(\mathbf{x}_k))^2}[\nabla^2 f(\mathbf{x}_k)]^{-1}$.

*Proof.* Consider the case *(i)*. From the upper bound in equation 24 that arises for self-concordant functions we have that

$$\nabla^2 f(\mathbf{x}_{k+1}) \preceq \frac{1}{(1-t_k\|\hat{\mathbf{d}}_{h,k}\|_{\mathbf{x}_k})^2}\nabla^2 f(\mathbf{x}_k)$$
$$\preceq \frac{1}{(1-t_k\hat{\lambda}(\mathbf{x}_k))^2}\nabla^2 f(\mathbf{x}_k).$$

where the last inequality holds from equation 35. As for the case *(ii)*, we make use of the lower bound in equation 24, and thus, by equation 35, we have

$$\nabla^2 f(\mathbf{x}_{k+1}) \succeq (1-t_k\hat{\lambda}(\mathbf{x}_k))^2 \nabla^2 f(\mathbf{x}_k).$$

Since, further, $f$ is strictly convex we take

$$[\nabla^2 f(\mathbf{x}_{k+1})]^{-1} \preceq \frac{1}{(1-t_k\hat{\lambda}(\mathbf{x}_k))^2}[\nabla^2 f(\mathbf{x}_k)]^{-1},$$

which concludes the proof. □

The next lemma estimates the sub-optimality gap of the process.

**Lemma C.4.** *Let* $\lambda(\mathbf{x}_k) < 1$. *Then,*

$$\omega(\lambda(\mathbf{x}_k)) \le f(\mathbf{x}_k) - f(\mathbf{x}^*) \le \omega_*(\lambda(\mathbf{x}_k)).$$

*If in addition* $\lambda(\mathbf{x}_k) < 0.68$ *then,*

$$f(\mathbf{x}_k) - f(\mathbf{x}^*) \le \lambda(\mathbf{x}_k)^2.$$

*Proof.* The first result follows directly from Lemma A.2 Further, it holds that $\omega_*(x) \le x^2, 0 \le x \le 0.68$, and thus we take the second result. □

Thus $\lambda(\mathbf{x}_k)$ can be used as an exit condition for Algorithm 1. Combining the result above with equation 38 one may take

$$f(\mathbf{x}_k) - f(\mathbf{x}^*) \le \omega_*\left(\frac{\sigma_{k,p+1}^{1/2}}{\mu_k\sigma_{k,N}^{1/2}}\hat{\lambda}(\mathbf{x}_k)\right)$$

and thus, for $\hat{\lambda}(\mathbf{x}_k) < \mu_k\sqrt{\frac{\sigma_{k,N}}{\sigma_{k,p+1}}}$, $\hat{\lambda}(\mathbf{x}_k)$ can be used as an exit condition for Algorithm 1.

**Lemma C.5.** *Let* $t_k := \frac{1}{1+\hat{\lambda}(\mathbf{x}_k)}$. *Then, for any* $k \in \mathbb{N}$ *we have*

$$f(\mathbf{x}_{k+1}) - f(\mathbf{x}_k) \le -\omega(\hat{\lambda}(\mathbf{x}))$$

*where* $\omega(x)$ *is defined in equation 22.*

*Proof.* By Lemma A.1 we have that

$$f(\mathbf{x}_k + t_k\hat{\mathbf{d}}_{h,k}) \le f(\mathbf{x}_k) + \langle\nabla f(\mathbf{x}_k), \mathbf{x}_{k+1} - \mathbf{x}_k\rangle + \omega_*(\|\mathbf{x}_{k+1} - \mathbf{x}_k\|_{\mathbf{x}_k})$$
$$= f(\mathbf{x}_k) + t_k\nabla f(\mathbf{x}_k)^T\hat{\mathbf{d}}_{h,k} + \omega_*(t_k\|\hat{\mathbf{d}}_{h,k}\|_{\mathbf{x}_k})$$
$$\le f(\mathbf{x}_k) - t_k\hat{\lambda}(\mathbf{x})^2 + \omega_*(t_k\hat{\lambda}(\mathbf{x}))$$

where for the last inequality we used equation 35, the monotonicity of $\omega_*(x)$ and $\hat{\lambda}(\mathbf{x})^2 = -\nabla f(\mathbf{x}_k)^T \hat{\mathbf{d}}_{h,k}$. Then, using the definition of $t_k$ we take

$$f(\mathbf{x}_k + t_k \hat{\mathbf{d}}_{h,k}) \leq f(\mathbf{x}_k) - \frac{\hat{\lambda}(\mathbf{x}_k)^2}{1 + \hat{\lambda}(\mathbf{x}_{h,k})} + \omega_* \left( \frac{\hat{\lambda}(\mathbf{x}_k)}{1 + \hat{\lambda}(\mathbf{x}_{h,k})} \right)$$
$$= f(\mathbf{x}_k) - \hat{\lambda}(\mathbf{x}_k) + \log \left( 1 + \hat{\lambda}(\mathbf{x}_k) \right).$$

which concludes the proof. $\qquad \square$

**Lemma C.6.** *Let $\lambda(\mathbf{x}_k) < 1$ and set $t_{h,k} = 1$. Further, suppose that assumption 2 holds. Then*

$$\lambda(\mathbf{x}_{k+1}) \leq \frac{\sqrt{1 - \bar{\varepsilon}_k \mu_k^2} + \lambda(\mathbf{x}_k)}{(1 - \lambda(\mathbf{x}_k))^2} \lambda(\mathbf{x}_k),$$

*with probability $\delta$.*

*Proof.* By Lemma C.3, inequality equation 37 and $t_k = 1$ we have

$$\nabla^2 f(\mathbf{x}_{k+1}) \preceq \frac{1}{(1 - \lambda(\mathbf{x}_k))^2} \nabla^2 f(\mathbf{x}_k), \qquad (39)$$

with probability one. Now, by the definition of the Newton decrement we have that

$$\lambda(\mathbf{x}_{k+1}) = \left[ \nabla f(\mathbf{x}_{k+1})^T [\nabla^2 f(\mathbf{x}_{k+1})]^{-1} \nabla f(\mathbf{x}_{k+1}) \right]^{1/2},$$

and combining this with inequality equation 39 we take

$$\lambda(\mathbf{x}_{k+1}) \leq \frac{1}{1 - \lambda(\mathbf{x}_k)} \left\| [\nabla^2 f(\mathbf{x}_k)]^{-1/2} \nabla f(\mathbf{x}_{k+1}) \right\|_2. \qquad (40)$$

Denote $\mathbf{Z} := [\nabla^2 f(\mathbf{x}_k)]^{-1/2} \nabla f(\mathbf{x}_{k+1})$. Using the fact that

$$\nabla f(\mathbf{x}_{k+1}) = \int_0^1 \nabla^2 f(\mathbf{x}_k + y \hat{\mathbf{d}}_{h,k}) \hat{\mathbf{d}}_{h,k} \, dy + \nabla f(\mathbf{x}_k)$$

we see that

$$\mathbf{Z} = [\nabla^2 f(\mathbf{x}_k)]^{-1/2} \left( \int_0^1 \nabla^2 f(\mathbf{x}_k + y \hat{\mathbf{d}}_{h,k}) \hat{\mathbf{d}}_{h,k} \, dy + \nabla f(\mathbf{x}_k) \right)$$
$$= \underbrace{\int_0^1 [\nabla^2 f(\mathbf{x}_k)]^{-1/2} \nabla^2 f(\mathbf{x}_k + y \hat{\mathbf{d}}_{h,k}) [\nabla^2 f(\mathbf{x}_k)]^{-1/2} \, dy}_{\mathbf{T}} \, [\nabla^2 f(\mathbf{x}_k)]^{1/2} \hat{\mathbf{d}}_{h,k} - [\nabla^2 f(\mathbf{x}_k)]^{1/2} \mathbf{d}_{h,k},$$

where $\mathbf{d}_{h,k}$ is the Newton direction. Next adding and subtracting the quantity $\mathbf{T}[\nabla^2 f(\mathbf{x}_k)]^{1/2} \mathbf{d}_{h,k}$ we have that

$$\mathbf{Z} = \mathbf{T}[\nabla^2 f(\mathbf{x}_k)]^{1/2} (\hat{\mathbf{d}}_{h,k} - \mathbf{d}_{h,k}) + (\mathbf{T} - \mathbf{I}_{N \times N})[\nabla^2 f(\mathbf{x}_k)]^{1/2} \mathbf{d}_{h,k},$$

and thus

$$\|\mathbf{Z}\| \leq \left\| \underbrace{\mathbf{T}[\nabla^2 f(\mathbf{x}_k)]^{1/2} (\hat{\mathbf{d}}_{h,k} - \mathbf{d}_{h,k})}_{\mathbf{Z}_1} \right\|_2 + \left\| \underbrace{(\mathbf{T} - \mathbf{I}_{N \times N})[\nabla^2 f(\mathbf{x}_k)]^{1/2} \mathbf{d}_{h,k}}_{\mathbf{Z}_2} \right\|_2 \qquad (41)$$

Using again equation 39 and since, by assumption, $y\hat{\lambda}(\mathbf{x}_k) < 1$ we take

$$[\nabla^2 f(\mathbf{x}_k)]^{-1/2} \nabla^2 f(\mathbf{x}_k + y \hat{\mathbf{d}}_{h,k}) [\nabla^2 f(\mathbf{x}_k)]^{-1/2} \preceq \frac{1}{(1 - y\lambda(\mathbf{x}_k))^2} \mathbf{I}_{N \times N}.$$

We are now in position to estimate both norms in equation 41. For the first one we have that

$$\|\mathbf{Z}_1\| \leq \left\| \int_0^1 \frac{1}{(1 - y\lambda(\mathbf{x}_k))^2} dy \right\|_2 \left\| [\nabla^2 f(\mathbf{x}_k)]^{1/2} \left( \hat{\mathbf{d}}_{h,k} - \mathbf{d}_{h,k} \right) \right\|_2$$

$$= \frac{1}{1 - \lambda(\mathbf{x}_k)} \left\| [\nabla^2 f(\mathbf{x}_k)]^{1/2} \left( \hat{\mathbf{d}}_{h,k} - \mathbf{d}_{h,k} \right) \right\|_2$$

$$\leq \frac{\sqrt{\lambda(\mathbf{x}_k)^2 - \hat{\lambda}(\mathbf{x}_k)^2}}{1 - \lambda(\mathbf{x}_{h,k})}.$$

Next, the second norm implies

$$\|\mathbf{Z}_2\| \leq \left\| \int_0^1 \left( \frac{1}{(1 - y\lambda(\mathbf{x}_k))^2} - 1 \right) dy \right\|_2 \left\| [\nabla^2 f(\mathbf{x}_k)]^{1/2} \mathbf{d}_{h,k} \right\|_2$$

$$= \frac{\lambda(\mathbf{x}_k)}{1 - \lambda(\mathbf{x}_k)} \lambda(\mathbf{x}_k).$$

Putting this all together, inequality equation 40 becomes

$$\lambda(\mathbf{x}_{k+1}) \leq \frac{\sqrt{\lambda(\mathbf{x}_k)^2 - \hat{\lambda}(\mathbf{x}_k)^2}}{(1 - \lambda(\mathbf{x}_k))^2} + \frac{\lambda(\mathbf{x}_k)}{(1 - \lambda(\mathbf{x}_k))^2} \lambda(\mathbf{x}_k).$$

Therefore, by lemma C.2 we take

$$\lambda(\mathbf{x}_{k+1}) \leq \left( \frac{\sqrt{1 - \bar{\varepsilon}_k \mu_k^2}}{(1 - \lambda(\mathbf{x}_k))^2} + \frac{\lambda(\mathbf{x}_k)}{(1 - \lambda(\mathbf{x}_k))^2} \right) \lambda(\mathbf{x}_k),$$

with probability $\delta$. □

*Proof of theorem 3.2.* From eq. (38) we have that

$$\mu_{\min} \sqrt{\bar{\varepsilon}_{\min}} \lambda(\mathbf{x}_k) \leq \mu_k \sqrt{\frac{\sigma_{k,N}}{\sigma_{k,p+1}}} \lambda(\mathbf{x}_k) \leq \hat{\lambda}(\mathbf{x}_k),$$

with probability $\delta$ for each $k \in \mathbb{N}$. Using this inequality, the result lemma C.5 and the monotonicity of $\omega$ we take

$$f(\mathbf{x}_{k+1}) - f(\mathbf{x}_k) \leq -\omega \left( \mu_{\min} \sqrt{\bar{\varepsilon}_{\min}} \lambda(\mathbf{x}) \right) \leq -\omega \left( \mu_{\min} \sqrt{\bar{\varepsilon}_{\min}} \hat{\eta} \right),$$

where the second inequality uses the assumption $\lambda(\mathbf{x}) > \hat{\eta}$. The above bound holds with probability $\delta$, and, further, $\gamma := \omega \left( \mu_{\min} \sqrt{\bar{\varepsilon}_{\min}} \hat{\eta} \right) > 0$ as required.

Additionally, if $\lambda(\mathbf{x}) \leq \hat{\eta}$, then from lemma C.6 we get that

$$\lambda(\mathbf{x}_{k+1}) \leq \frac{\sqrt{1 - \bar{\varepsilon}_{\min} \mu_{\min}^2} + \lambda(\mathbf{x}_k)}{(1 - \lambda(\mathbf{x}_k))^2} \lambda(\mathbf{x}_k) \leq \frac{\sqrt{1 - \bar{\varepsilon}_{\min} \mu_{\min}^2} + \hat{\eta}}{(1 - \hat{\eta})^2} \lambda(\mathbf{x}_k),$$

with probability $\delta$. Moreover, let $\bar{\varepsilon} \in (0,1)$ and $\mu \in (0,1)$. Then $\bar{\varepsilon}_{\min}, \mu \in (0,1]$. By $\hat{\eta} := \frac{3 - \sqrt{5 + 4\hat{\varepsilon}}}{2}$, we have that $\hat{\eta} \in (0, \frac{3 - \sqrt{5}}{2})$. Then, from the definition of $\hat{\eta}$, it holds

$$\frac{\sqrt{1 - \bar{\varepsilon}_{\min} \mu_{\min}^2} + \hat{\eta}}{(1 - \hat{\eta})^2} < 1,$$

and thus $\lambda(\mathbf{x}_{k+1}) < \lambda(\mathbf{x}_k)$. Therefore, this process converges with a composite rate and a probability $\delta$ at each $k \in \mathbb{N}$.

If, on the other hand, $\bar\varepsilon = \mu = 1$, then from lemma C.6 we have that

$$\lambda(\mathbf{x}_{k+1}) \leq \frac{\sqrt{1 - \bar\varepsilon_k \mu_k^2} + \hat\eta}{(1 - \hat\eta)^2} \lambda(\mathbf{x}_k),$$

with probability $\delta$. Since $\sqrt{1 - \bar\varepsilon_k \mu_k^2} \in (0, 1)$ and using the definition of $\hat\eta$ as above, we obtain $\lambda(\mathbf{x}_{k+1}) < \lambda(\mathbf{x}_k)$. Hence, since $\bar\varepsilon = \mu = 1$, then both $(\bar\varepsilon_k)_{k \in \mathbb{N}}$ and $(\mu_k)_{k \in \mathbb{N}}$ converge to 1 and thus

$$\frac{\lambda(\mathbf{x}_{k+1})}{\lambda(\mathbf{x}_k)} \leq \frac{\sqrt{1 - \bar\varepsilon_k \mu_k^2} + \lambda(\mathbf{x}_k)}{(1 - \lambda(\mathbf{x}_k))^2} \quad \text{and} \quad \lim_{k \to \infty} \frac{\sqrt{1 - \bar\varepsilon_k \mu_k^2} + \lambda(\mathbf{x}_k)}{(1 - \lambda(\mathbf{x}_k))^2} = 0.$$

Therefore, we conclude that the process converges with a super-linear rate and probability $\delta$ at each $k \in \mathbb{N}$. $\qquad\square$

# D   Proofs of Results of Section 3.3

*Proof of Theorem 3.3.* Let $\hat\lambda(\mathbf{x})$ be the the the approximate decrement based on the current coarse model

$$\hat\lambda(\mathbf{x}_k) := \left[ \nabla f(\mathbf{x}_k)^T \mathbf{P}_k |\mathbf{Q}_{H,k}^{-1}| \mathbf{R}_k \nabla f(\mathbf{x}_k) \right]^{1/2},$$

$|\mathbf{Q}_{H,k}^{-1}|$ are defined in equation 16. The following results will be used later in the proof. By construction of $|\mathbf{Q}_{H,k}^{-1}|$ and assumption 3 for all $k \in \mathbb{N}$ we have

$$\nu \mathbf{I}_n \preceq |\sigma_{k,p+1}| \mathbf{I}_n \preceq |\mathbf{Q}_{H,k}| \preceq M \mathbf{I}_n,$$

where the left-hand side inequality holds by (11). This implies

$$\|\hat{\mathbf{d}}_{h,k}\|^2 = \|\mathbf{P}_k |\mathbf{Q}_{H,k}^{-1}| \mathbf{R}_k \nabla f(\mathbf{x}_k)\| \leq \frac{\omega^2}{\nu} \hat\lambda(\mathbf{x}_k)^2, \tag{42}$$

where, here, for simplicity, we use $\hat{\mathbf{d}}_{h,k}$ instead of $|\hat{\mathbf{d}}_{h,k}|$ in equation 17. We also obtain

$$\hat\lambda(\mathbf{x}_k)^2 \geq \frac{1}{M} \|\mathbf{R}_k \nabla f(\mathbf{x}_k)\|^2 \geq \frac{\hat\mu^2}{M} \|\nabla f(\mathbf{x}_k)\|^2, \tag{43}$$

with probability $\delta > 0$. Furthermore, by assumption 3 we take

$$\begin{aligned}
f(\mathbf{x}_{k+1}) - f(\mathbf{x}_k) &\leq -t_k \hat\lambda(\mathbf{x}_k)^2 + \frac{M}{2} t_k^2 \|\hat{\mathbf{d}}_{h,k}\|_2^2 \\
&\overset{(43)}{\leq} -t_k \left( 1 - \frac{M\omega^2}{2\nu} t_k \right) \hat\lambda(\mathbf{x}_k)^2 \\
&\leq -\frac{\nu}{2M\omega^2} \hat\lambda(\mathbf{x}_k)^2,
\end{aligned}$$

where the last inequality follows from our assumption to $t_k$. Finally, using eq. (43) into the last inequality we get the desired result with probability $\delta > 0$. $\qquad\square$

*Proof of Theorem 3.4.* Combining the result of theorem 3.3 and that for all $k \in \mathbb{N}$ we have that $|\sigma_{k,p+1}| \geq \nu$ we obtain

$$f(\mathbf{x}_{k+1}) - f(\mathbf{x}_k) \leq -\frac{\hat\mu^2 \nu}{2\omega^2 M} \|\nabla f(\mathbf{x}_k)\|^2,$$

with probability $\delta$. Using assumption 5 we get

$$f(\mathbf{x}_{k+1}) - f(\mathbf{x}_k) \leq -\frac{\hat\mu^2 \nu \xi}{\omega^2 M} (f(\mathbf{x}_k) - f(\mathbf{x}^*)).$$

The result follows by adding and subtracting $f(\mathbf{x}^*)$ in the above inequality. $\qquad\square$

| Datasets | Problem | $m$ | $n$ | $N$ | $p$ |
|----------|---------|-----|-----|-----|-----|
| CTslices | Non-linear least-squares | $53,500$ | $385$ | $0.5n$ | $60$ |
| CovType | Non-linear least-squares | $581,012$ | $54$ | $0.7n$ | $5$ |
| Gisette | Non-linear least-squares | $6,000$ | $5,000$ | $0.5n$ | $350$ |
| W8T | Non-linear least-squares | $14,951$ | $300$ | $0.5n$ | $60$ |

Table 2: Set-up and dataset details for non-convex, non-linear regression problem.

# E   Extra Numerical Results and Details

All the datasets used in the experiments can be found in `https://www.csie.ntu.edu.tw/~cjlin/libsvmtools/datasets/` and `http://archive.ics.uci.edu/ml/index.php`. What follows is a description of the algorithms used in comparisons.

1. Gradient Descent (GD) with Armijo-rule

2. Accelerated Gradient Descent (AccGD) with Armijo-rule and momentum 0.5.

3. NewSamp with Armijo-rule (Erdogdu & Montanari, 2015)

4. Adam with $t_k = 10^{-3}, \beta_1 = 0.9, \beta_2 = 0.99, \epsilon = 10^{-8}$ (Kingma & Ba, 2014)

5. Sigma with Armijo-rule (Tsipinakis & Parpas, 2024)

6. SigmaSVD with Armijo-rule

7. Cubic Regularization of the Newton's method (Cubic Newton) with line search (Nesterov et al., 2018)

To be fair in the comparisons we perform the same step size strategy for all algorithms but Adam. Note that NewSamp forms the Hessian matrix as in equation 10 and although it is not directly comparable to our approach, we include it in our experiments to show that our approach outperforms sub-sampled Newton methods. Similar to $N$, we denote $|S_m|$ the number of samples that NewSamp uses to form the Hessian. In addition, the number $p$ of the eigenvalues is selected the same for both SigmaSVD and NewSamp in all the experiments. The results of this section were generated on standard laptop machine with Intel i7-10750H CPU processor. The numerical results of main text were generated on GPU processor in Google's colab. The code for generating all the numerical results will be uploaded on github in the near future.

## E.1   Non-linear regression

In this section we revisit the non-convex problem of Section 4 to perform simulations on different datasets. Full details on the datasets used for the non-linear least-squares problem and how the parameters $N$ and $p$ are selected are given in Table 2. Figure 3 shows that SigmaSVD, Adam and Cubic Newton are the best algorithms as they return the lowest training errors, yet SigmaSVD does better in problems with several saddle points or flat areas. The three algorithms perform similar in terms of GPU time when the problem dimensions is small. However, for problems with $n$ large the results are favourable for SigmaSVD.

## E.2   Log-linear regression

Here we minimize the following self-concordant function,

$$\mathbf{x}^* = \arg\min_{\mathbf{x}\in\mathbb{R}^n} f(\mathbf{x}) = \arg\min_{\mathbf{x}\in\mathbb{R}^n} \left( -\sum_{i=1}^{m} \log(b_i - \mathbf{a}_i^T \mathbf{x}) \right).$$

We generate two datasets to illustrate the efficiency of the proposed method for the regimes $m > n$ and $n > m$ as follows: $\{\mathbf{a_i}\}_{i=1}^m$ is generated from the multivariate Gaussian distribution with zero mean and unit

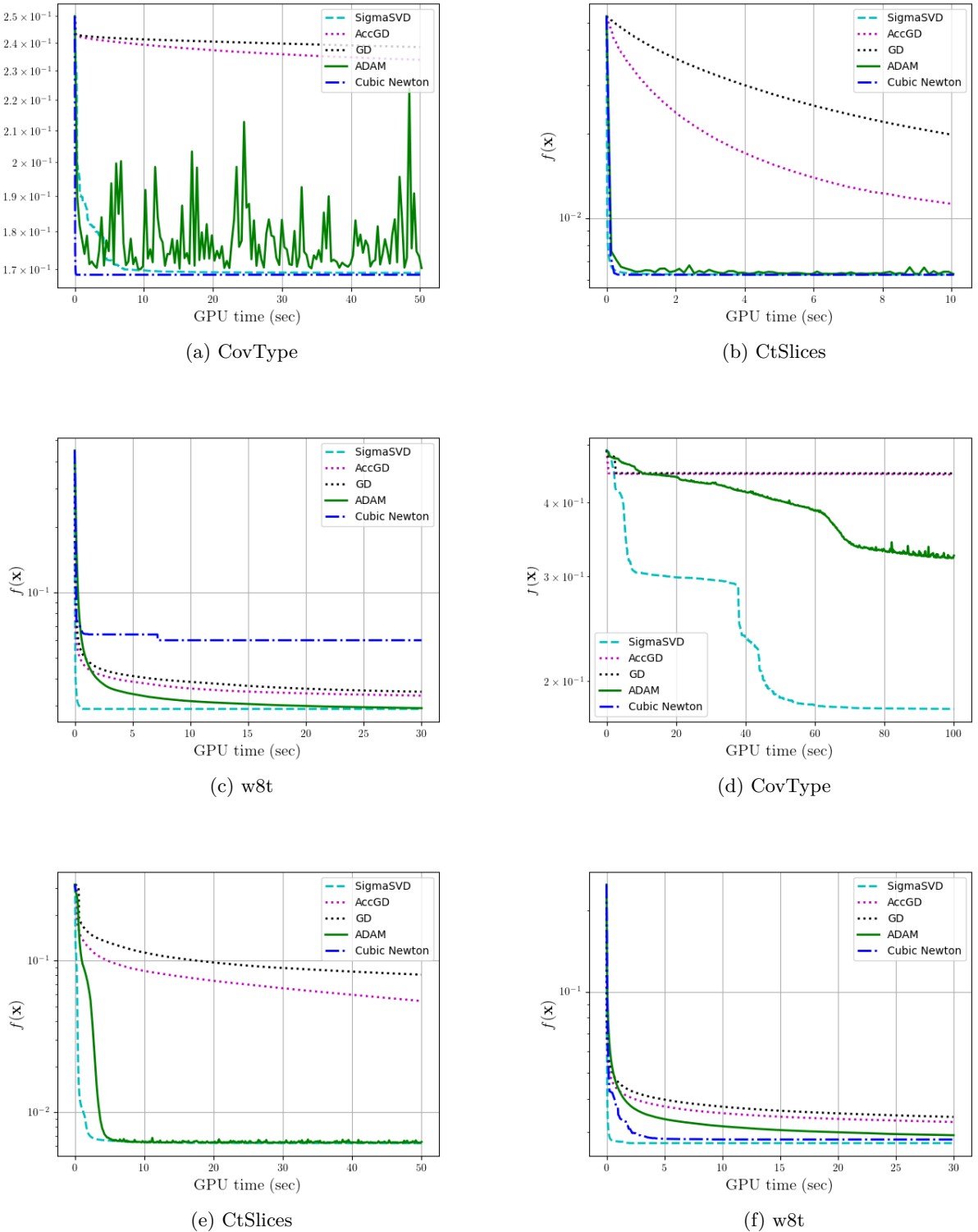

Figure 3: Non-convex minimization. All methods in plots from (a) to (c) are initialized at the origin while from (d) to (f) the initializer is selected randomly from a Gaussian $\mathcal{N}(0,1)$.

| Datasets | Problem | $m$ | $n$ | $N$ | $|S_m|$ | $p$ |
|---|---|---|---|---|---|---|
| Synthetic | Log linear model | $10,000$ | $1,000$ | $0.5n$ | $0.3m$ | $150$ |
| Synthetic | Log linear | $1,000$ | $10,000$ | $0.1n$ | $-$ | $150$ |

Table 3: Set-up and dataset details for Log-linear regression.

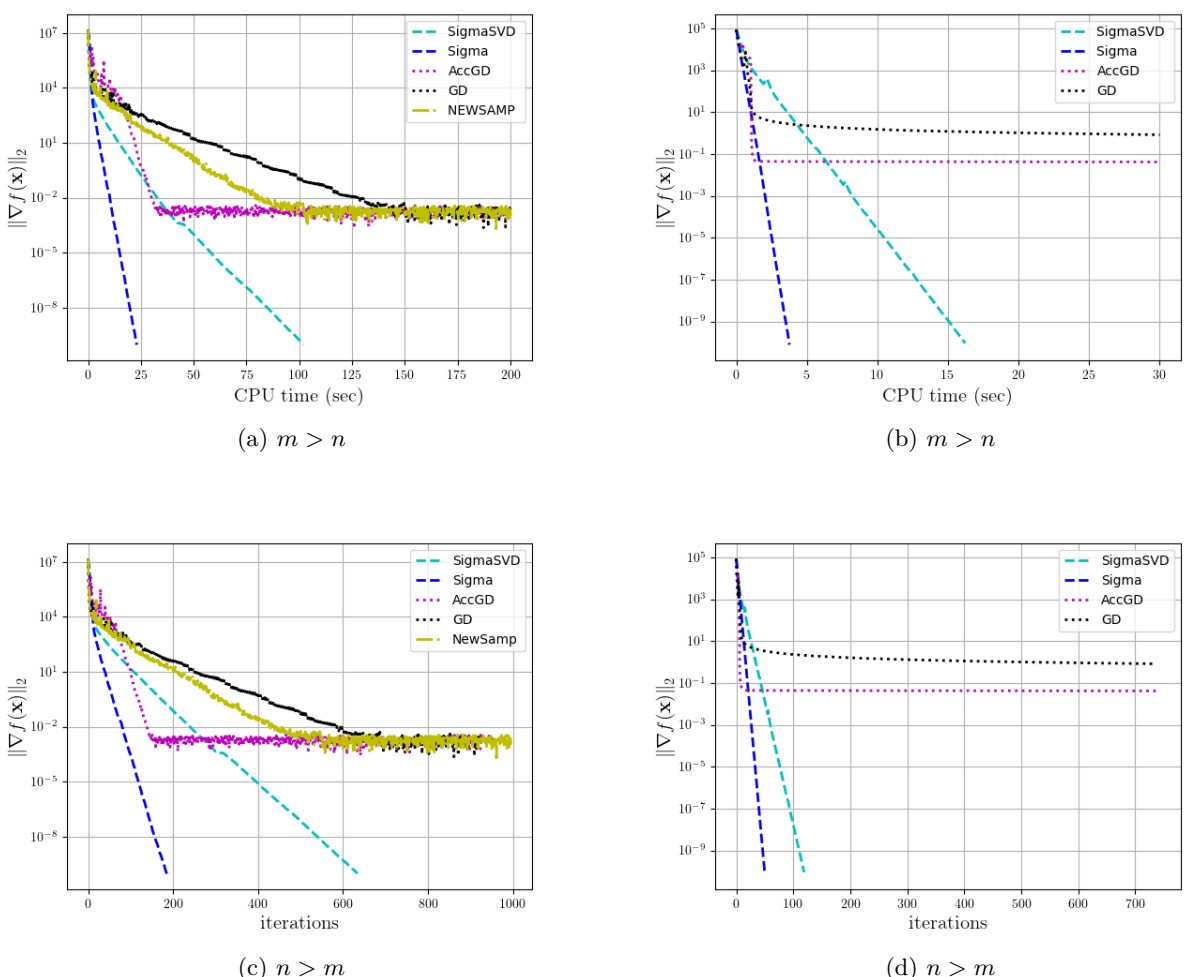

(a) $m > n$

(b) $m > n$

(c) $n > m$

(d) $n > m$

Figure 4: Log Linear Regression. Plots (a) and (c) show comparisons between the optimization algorithms for the regime $m > n$ while (b) and (d) for the regime $m < n$.

variance and $\{b_i\}_{i=1}^m$ from uniform distribution. Full details and the set-up for algorithm 1 are given in Table 3.

Figure 4 shows the performance of the optimization algorithms. Due to the domain of this problem, the performance of Adam is missing. The performance of NewSamp is missing from the experiment for the regime $n > m$ due to the large value of $n$. Clearly, the fastest method is Sigma which achieves a very fast super-linear convergence rate. As shown in Theorem 3.2, the reason for the difference in the performance between Sigma and SigmaSVD is that the ratio $\sigma_{p+1}/\sigma_n$ is large which indicates that performing SVD on the reduced Hessian matrix we discard much of the important second-order information of the problem. However, even in such a scenario, in contrast to NewSamp and gradient descent methods, SigmaSVD is able to converge to the global minimum with a super-linear rate.

| Datasets | Problem | $m$ | $n$ | $N$ | $|S_m|$ | $p$ | $\ell$ |
|---|---|---|---|---|---|---|---|
| CTslices | Logistic model | $53,500$ | $385$ | $0.5n$ | $0.3m$ | $60$ | $10^{-6}$ |
| Leukemia | Logistic model | $38$ | $7,129$ | $0.1n$ | $0.7m$ | $150$ | $10^{-6}$ |
| News20 | Logistic model | $19,996$ | $1,355,129$ | $0.001n$ | $-$ | $100$ | $10^{-12}$ |

Table 4: Set-up and dataset details for Logistic regression.

| Datasets | Problem | $m$ | $n$ | $N$ | $|S_m|$ | $p$ | $\ell$ |
|---|---|---|---|---|---|---|---|
| CovType | SVM | $581,012$ | $54$ | $0.5n$ | $0.3m$ | $10$ | $10^{-2}$ |
| W8T | SVM | $14,951$ | $300$ | $0.5n$ | $0.5m$ | $60$ | $10^3$ |

Table 5: Set-up and dataset details for Support Vector Machines.

### E.3 Logistic regression

In this section we are concerned with the problem of finding the maximum likelihood in generalized linear models. In particular, we consider the regularized logistic regression which as an optimization problem takes the following form,

$$\mathbf{x}^* = \arg\min_{\mathbf{x} \in \mathbb{R}^n} f(\mathbf{x}) = \arg\min_{\mathbf{x} \in \mathbb{R}^n} \left( \frac{1}{m} \sum_{i=1}^m \log(1 + e^{-b_i \mathbf{a}_i^T \mathbf{x}_h}) + \frac{\ell}{2} \|\mathbf{x}\|_2^2 \right).$$

Note that the logistic model is a strongly convex function with Lipschitz continuous Hessian matrix. The details and results for the logistic regression experiment are given in Table 4 and Figure 5, respectively. Note that for leukemia dataset, $m$ is too small to perform batch learning and thus the performance of Adam is omitted for this example. Second-order methods clearly outperform first-order methods for the CtSlices dataset. Further, NewSamp is slightly faster than the multilevel methods but obtains slightly worse convergence rate (Figures 5a) and 5d, respectively). However, as expected, the efficiency of NewSamp reduces drastically for large values of $n$ (Leukemia dataset). Note that in this example the multilevel methods perform similarly which indicates that there is no need to use the full spectrum of the reduced Hessian matrix. Last, Figures 5c and 5f show the efficiency of SigmaSVD on a problem with over a million parameters. Such a problem lies at the heart of large scale machine learning and deep learning. Even when the coarse dimensions are selected very small ($N = 0.001n$ in this experiment), Figure 5c shows that multilevel methods are capable of returning solution with very good accuracy much faster than first-order methods.

### E.4 Support Vector Machines

We can train Support Vector Machines (SVMs) using the primal problem over *hinge-q loss* or *Huber loss* function. Since our approach requires twice differentiable functions we consider the *hinge-2 loss* function for training the SVMs,

$$\mathbf{x}^* = \arg\min_{\mathbf{x} \in \mathbb{R}^n} \frac{1}{2} \|\mathbf{x}\|_2^2 + \frac{\ell}{2} \sum_{i=1}^m \max\{0, 1 - b_i \mathbf{a}_i^T \mathbf{x}\}^2.$$

Note that the objective function of this section is convex, however the Hessian matrix is not Lipschitz continuous. The details and results for the logistic regression experiment are given in Table 5 and Figure 6, respectively. Again, both multilevel methods outperform its counterparts. Here, as also in the logistic regression example, using only a few eigenvalues to form the reduced Hessian matrix does not seem to affect the performance of SigmaSVD. Further, we observed that the performance of NewSamp becomes very poor, even compared to first-order methods, when highly regularized solutions are required (Figures 6b and 6d). On the other hand, both multilevel methods, no matter how large or small the regularization parameter is chosen, always outperform the first-order methods.

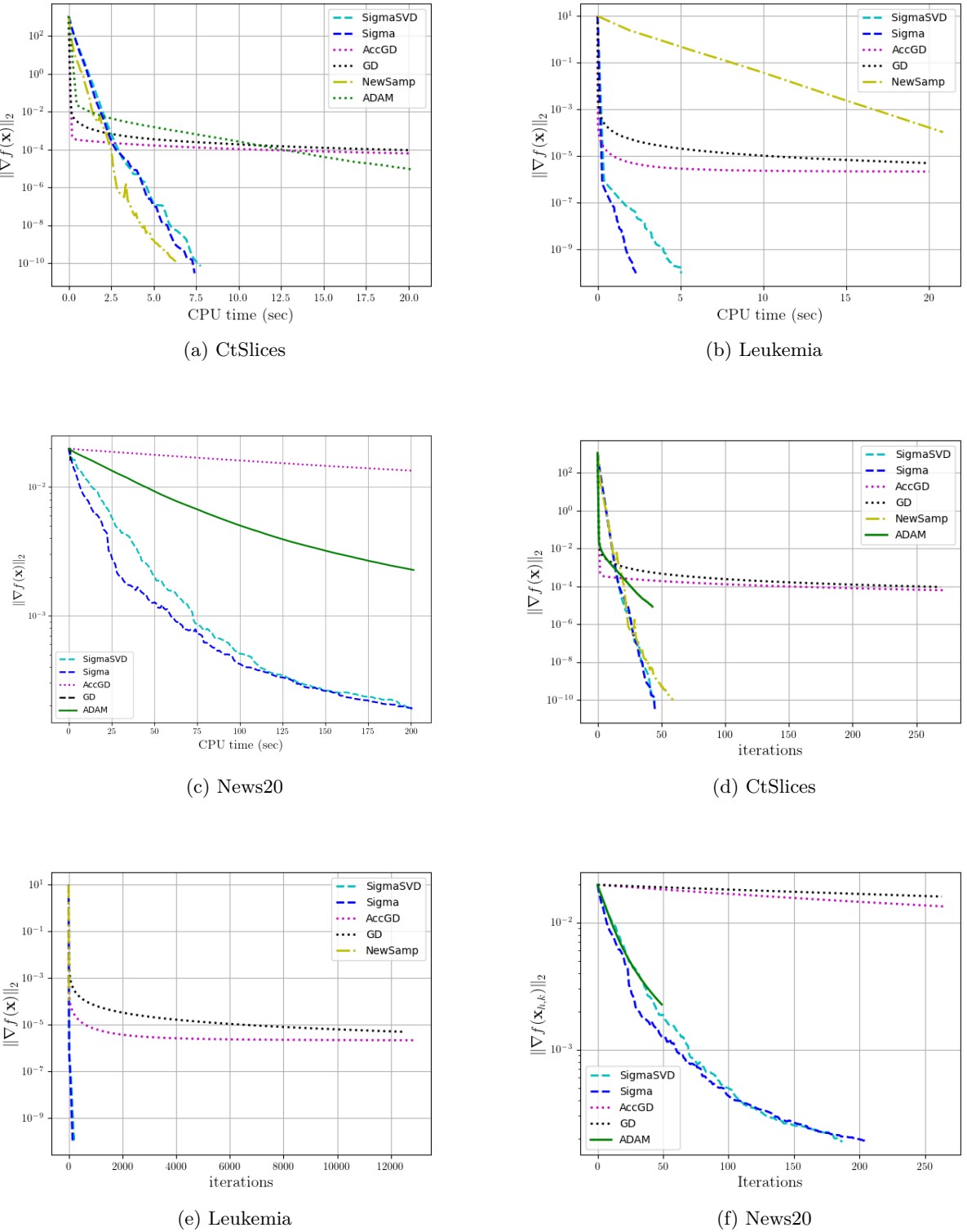

Figure 5: Logistic Regression. Plots (a) to (c) show the norm of the gradient vs cpu time in seconds while (d) to (f) show the norm of the gradient vs iterations for three machine learning datasets.

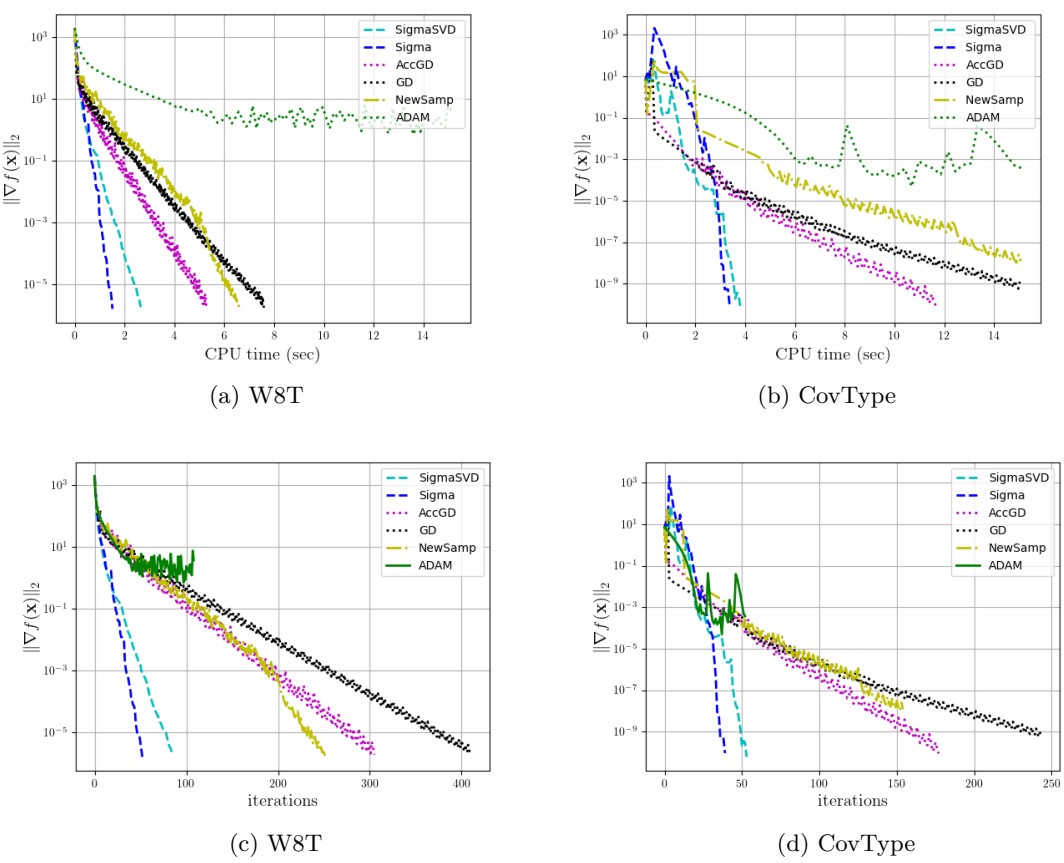

Figure 6: Support Vector Machines. Plots (a) and (b) show the norm of the gradient vs cpu time in seconds while (c) and (d) norm of the gradient vs iterations for two machine learning datasets.

