# OpenReview forum: "A Multilevel Low-Rank Newton Method with Super-linear Convergence Rate and its Application to Non-convex Problems"
_TMLR — Accepted by TMLR_

### Review · Reviewer_ATKW · 2025-10-29

**Summary Of Contributions:**

This paper presents a reduced-rank multi-level Newton's method for convex and non-convex objectives. Some simple theoretical properties of the presented methods are established, and numerical results are obtained on MNIST data sets.

**Additional Comments:**

N/A

**Audience:**

No

**Audience Explanation:**

At this moment, I feel both theoretical and numerical results are too limited to be of broad interest - however, with changes made (see below), the paper has the potential to reach many ML/optimization researchers for its relevance.

**Claims And Evidence:**

Yes

**Claims Explanation:**

descriptions of methods, and imposed assumptions are accurate, proper, and numerical results seem reasonable.

**Requested Changes:**

I have two major comments about results in this paper.

1. On the convex side, the paper assumes the objective function is self-concordant. While this is (somewhat) a standard assumption for Newton's method, I find it a bit concerning: usually, self-concordant functions arise as barrier functions of convex constraints, which are then optimized (using Newton's method) as an intermediate step in interior point methods. In this scenario, it is easy to verify self concordance because it is by construction of barrier functions. In this paper, however, it is directly assumed that the objective function is self concordant: how do we know this is the case? Are common loss functions in ML, e.g. least squares/cross entropy, self-concordant functions? If so, what are their self-concordant parameters (functions of dimension n/N) and how are they related to the properties of the training data?

The authors must clarify this, preferably with real-world examples in machine learning or statistics, to justify the relevance of proposed method which requires self-concordance of the objective function.

2. On the non-convex side, I have a few comments.

2-(a). I do not find a similar assumption to self concordance when the function is non-convex. The authors state the PL condition, which is well-known to be similar to strong convexity and sufficient for the purpose of first-order methods, but how about self concordance? Without self concordance, is Newton's method very useful, compared simply with fixed step size first-order methods? This also leads me to my second question...

2-(b). The convergence rates established in Theorem 3.4 are linear convergence, not quadratic convergence. Then why should we care about Newton's method? With PL condition, gradient descent achieves linear convergence. Even with stochastic gradients, variants of SVRG with finite sum structures also achieve linear convergence.

Overall, for my second criticism, I think quadratic convergence must be proved for non-convex functions to establish the relevance of Newton's method. If only linear convergence is obtained, then SVRG/Gradient Descent can already do that, with much smaller iteration complexity, on non-convex functions that satisfy the PL condition. If the argument is that Newton's method has a smaller constant in linear convergence, then it should be compared, theoretically, with constants from GD, SVRG, Accelerated GD, to see which one has smaller constants in which scenarios.

3. On the numerical side, MNIST/UCI might be too much a toy example to demonstrate the effectiveness of the proposed method. I recommend the following changes, to make the numerical experiments more interesting.

3-(a). Alongside with MNIST/UCI, the authors should study their methods on more synthetic data using simulations. I have a feelings that synthetic examples might be more interesting than MNIST, because by controlling various parameters of a testing function, different aspects (e.g. dependency on dimension, dependency on self-concordance parameter) could be revealed more clearly.

3-(b). In addition to MNIST/UCI, the authors should report performance of their methods on a more serious ML data set. I am not familiar with those, but I know MNIST is kind of too "toy".

---

> ### Author Response · Authors · 2025-12-24
>
> We thank Reviewer ATKW for their constructive feedback and for recognizing the potential of our work to reach a broad audience. We appreciate the opportunity to clarify the theoretical assumptions and the choice of experimental benchmarks. Below, we address the requested changes point-by-point.
>
> ## 1. On Self-Concordance in Unconstrained Optimization
>
> Self-concordance is an important assumption for Newton's method not only in constrained optimization (interior-point methods) but also in unconstrained optimization. It addresses an important shortcoming of the method under the classical assumptions of strong convexity and Lipschitz continuous Hessians, namely the lack of a scale-invariant analysis. Under self-concordance, the standard analysis of Newton's method becomes aligned with the Newton step itself, in the sense that both are invariant under affine transformations of the variables. This was not possible to achieve within the classical framework.
>
>
> The class of self-concordant functions includes $\mu$-strongly convex functions with $L$-Lipschitz continuous Hessians, as the latter are self-concordant with constant $M = L / (2\mu^{3/2})$. Therefore, if $f$ is three-times differentiable, the self-concordance assumption is less restrictive than the classical assumptions. We have added this discussion to the background section. For a detailed analysis of these claims, see [1].
>
>
> In our analysis, without loss of generality, we assume $M=1$ (standard self-concordant functions). Our results can immediately extend for any $M\ge0$.
>
> ## 2. Non-Convex Settings and Convergence Rates
>
> ## (a) Non-convex Self-concordance
> The framework of non-convex self-concordant functions is currently underexplored. We are only aware of one very recent paper dealing with non-convex self-concordant functions; see [2]. Thus, while this is a very interesting observation by the reviewer, such an analysis would be quite innovative and, in our view, more appropriate for a separate project that we plan to pursue in the future.
>
> ## (b) Linear vs. Quadratic Convergence under PL
> We interpret the reviewer's comment as a request for a "local" quadratic convergence result in the non-convex setting, since a global quadratic rate from an arbitrary initialization is impossible to establish in general. However, proving a local quadratic rate for our non-convex variant is not feasible, because our method truncates the true Hessian matrix. In such cases, one can at best expect a local composite or superlinear convergence rate, which is precisely what we establish in Theorems 3.1 and 3.2. We therefore believe that the issue raised by the reviewer is already addressed. Moreover, in the standard analysis of Newton's method for non-convex problems, a local quadratic rate essentially amounts to assuming that the iterates enter a strictly convex neighborhood of the minimizer, where the Hessian is locally positive definite. As a result, the term "local" in "local quadratic convergence" implicitly entails a strict convexity assumption, which is exactly the assumption we adopt in this paper, alongside the smoothness of the Hessian.
>
>
> Theorem 3.4 exhibits a linear convergence rate that is worse than that of gradient descent. This is expected, as the result is global, and, to the best of our knowledge, there is no global analysis of Newton's method in the literature that demonstrates improvements over first-order methods under the PL inequality or strong convexity. This issue is addressed in our recent paper [3], where we develop an adaptive multilevel method to alleviate this limitation. Moreover, to obtain theoretical improvements in both non-convex and convex settings, one typically needs to consider regularized Newton methods; see [1, Section 5]. We use this framework to establish such improvements over first-order methods in our recent paper [4]. Therefore, we believe that the additional developments requested by the reviewer are either already addressed or are not suitable to be fully treated within a single paper.
>
>
> The non-convex variant of our method is essentially a straightforward modification of the original SigmaSVD, and we illustrate that it performs well on practical non-convex problems. The theory in Section 3.3 is intended primarily to support this SigmaSVD variant. The main theoretical contributions of the paper lie in Sections 3.1 and 3.2, where we establish local composite and super-linear convergence rates for the proposed methods in convex settings---rates that are impossible to attain with first-order methods.

---

> > ### Author Response · Authors · 2025-12-24
> >
> > ## 3. Numerical Experiments and Dataset Difficulty
> >
> > We understand the reviewer's concern that MNIST is often viewed as a "toy" dataset in the context of simple classification. However, we specifically selected the MNIST Deep Autoencoder problem because it is a widely recognized benchmark for optimization difficulty, distinct from standard classification tasks. We agree that it is reasonable for the reviewer to question its difficulty: although we emphasize the difficulty of this problem, we neglected to cite relevant works that support this claim.
> >
> > The MNIST autoencoder problem has become a standard benchmark problem precisely because it is difficult to solve; see [5]. There, the author explicitly notes that, due to their high difficulty, performance on these problems has become a standard benchmark for neural network optimization methods (e.g., Martens and Grosse [2015], Sutskever et al. [2013], Vinyals and Povey [2012], Martens [2010]). The difficulty arises from the presence of saddle points and from the vanishing gradient phenomenon. In particular, the MNIST autoencoder problem considered here consists of $9$ layers in total and uses the sigmoid activation function. Deep networks with sigmoid activations are well known to suffer from vanishing gradients; see also [5] and reference therein. Thus, without a well-designed preconditioner, escaping such neighborhoods can be challenging for first-order methods.
> >
> > This is precisely why Adam slows down significantly in Figure 2. The fact that Adam struggles to escape such a region further supports the view that the MNIST autoencoder problem should not be regarded as a toy problem. In summary, we have applied our method to two difficult non-convex optimization problems that are known to pose significant challenges to optimization methods due to the presence of saddle points and flat regions. We are happy to evaluate the method on other difficult problems that the reviewer may suggest.
> >
> >
> > [1] Yurii Nesterov. Lectures on convex optimization, volume 137. Springer, 2018.
> >
> > [2] D. Goldfarb et al. Non-Convex Self-Concordant Functions: Practical Algorithms and Complexity Analysis, arXiv preprint arXiv:2511.15019, 2025.
> >
> > [3] N. Tsipinakis et al. Adaptive Multilevel Newton: A Quadratically Convergent Optimization Method, arXiv preprint arXiv:2510.24967, 2025.
> >
> > [4] N. Tsipinakis and P. Parpas. Multilevel Regularized Newton Methods with Fast Convergence Rates, SIAM Journal on Scientific Computing, S232-S257, 2025.
> >
> > [5] S. Reddi et al. A Generic Approach for Escaping Saddle points, International conference on artificial intelligence and statistics, 1233--1242, 2018.

---

### Review · Reviewer_VqxF · 2025-11-27

**Summary Of Contributions:**

The authors introduce the SigmaSVD method applied to smooth functions for unconstrained optimization. They show SigmaSVD to be superlinearly convergent in the convex setting under the assumption of self concordance and in the nonconvex setting under the assumption of self concordance and a PL assumption. The analysis is then extended to the nonconvex and stochastic setting where only a partial gradient and hessian vector product information is available. The proofs are done well, build off of previous work, and provide novel results.

The computational cost is shown to be less than a full Newton-like method, but the use of a truncated singular value decomposition (T-SVD) still requires considerably more computation than well implemented first order methods. Also, tuning how `coarse' (selecting $N<n$) the T-SVD should be is likely problem dependent. The authors do not discuss these complications in much detail.

When applied to a nonlinear least squares, SigmaSVD shows a considerable computational efficiency advantage. In this example, full gradient and hessian vector products are available. The second example applies SigmaSVD to training an autoencoder. Notably, in this example SigmaSVD is modified to incorporate some aspect of momentum inspired by ADAM. No substantive proofs are given for the algorithm under these modifications. Furthermore, a full SVD is used in this example. While the authors say this is reasonable given the size of the coarse model, the numerical results demonstrate a questionable computational advantage of SigmaSVD over ADAM.

**Audience:**

Yes

**Audience Explanation:**

While many of the building blocks of this algorithm are not particularly new, in conjunction the SigmaSVD algorithm shows considerable promise. The idea of extracting only enough curvature information to have a computational edge over gradient based methods appears to be a fruitful area of research and should be explored more.

**Claims And Evidence:**

Yes

**Claims Explanation:**

The background information, proofs and numerical experiments give clear evidence that SigmaSVD is superlinearly convergent as claimed. They demonstrate that the cost per iteration is considerably cheaper than other Newton-like variants.

**Requested Changes:**

My main critique is the lack of discussion surrounding how to choose the size of the T-SVD so that there still is an advantage over gradient based methods. Obviously, tuning was done to select $N$ for the examples given. It is not clear how sensitive the method's advantage is to the size of the T-SVD. Furthermore, one of the examples choose $N=n$ making the T-SVD a full SVD. This may indicate that truncating is not viable in practice.

---

> ### Author Response · Authors · 2025-12-24
>
> We thank Reviewer VqxF for their constructive feedback and for highlighting the potential of the SigmaSVD algorithm. We appreciate the opportunity to clarify the selection of the T-SVD size and the method's applicability range.
>
> Regarding the tuning of $N$ and the sensitivity of the method, we address these in the newly added Remark 3.1 and Remark 3.2 (please see our response to VAxm for details).
>
> In addition, it possible that T-SVD may not be always efficient to perform. This is related to the effective rank $R$ which depends on the problem. For example, for a linear least-squares problem with $m \gg n $, it is common to have $R = n$ and thus the method will not achieve the very fast rate locally. Thus, as we mention in the manuscript, the method should be applied to problems with low-rank Hessians--- such structures are particularly common for non-convex settings. We have made this clear in the introduction and conclusion.

---

### Review · Reviewer_VAxm · 2025-12-02

**Summary Of Contributions:**

The authors consider the numerical solution of large-scale optimization problems related to machine learning models. They introduce a novel variant of two-level Newton methods that relies on randomized methods for the computation of approximate second order information related to the reduced Hessian matrix. Exploiting such subspace information allows to reduce the global computational cost of the method. The resulting algorithm named SigmaSVD (Algorithm 1) is presented in Section 2 and analyzed at the theoretical level in Section 3 and Appendices B, C and D, respectively. A super-linear convergence rate is shown for such a low rank Newton method for self-concordant functions. The authors propose a variant applicable to non-convex problems and establish a linear rate of convergence when the Polyak-Lojasiewicz inequality is satisfied. Numerical experiments on various optimization problems related to machine learning are provided in Section 4 and Appendix E showing the efficiency and applicability of SigmaSVD.

Strengths
========
- The manuscript is well written and covers both theoretical and numerical aspects.

- Convincing numerical experiments given in Section 4 and Appendix E are provided. These numerical experiments support the claims related to the convergence rate and demonstrate the efficiency of the proposed algorithm. The ability of SigmaSVD to solve large-scale non-convex problems efficiently is also highlighted.

Weaknesses
============
- SigmaSVD relies on randomized methods for the computation of approximate second order information related to the reduced Hessian matrix. The authors employ the Nyström method with uniform sampling as a randomized method. Would it be possible to consider more advanced sampling methods that could offer a lower computational complexity for a similar accuracy? The authors are expected to provide more information and details on this aspect.

- The efficiency of the algorithm proposed by the authors is partly based on the fact that the reduced Hessian matrix exhibits a low rank structure for applications related to the optimization of machine learning models. This statement should be highlighted more clearly in the manuscript (e.g. in the Introduction and Conclusion). Reference articles related to this claim should be given and discussed.

- The efficiency of SigmaSVD is partly governed by two hyperparameters: the size of the coarse problem $N$ and the order of the low-rank approximation of the reduced Hessian matrix $p$. The influence of both parameters is shown at the experimental level only in Section 4 and Appendix E, respectively. Comments concerning the determination of the size of the coarse problem are expected at the theoretical level.

**Additional Comments:**

The authors should provide a link to their software repository. It is indeed important to help TMLR's audience to reproduce the numerical experiments provided in Section 4 and Appendix E, respectively. This would be a nice asset that may increase the impact of the manuscript.

**Audience:**

Yes

**Audience Explanation:**

The authors introduce a novel algorithm for the optimization of large-scale machine learning models. This algorithm is a second order method exhibiting a super linear convergencE rate and is also applicable when solving non-convex problems.

The manuscript offers two major contributions:
- a theoretical proof related to the convergence rate of the method (given in  Appendices B and C),
- detailed numerical experiments on different machine learning models that provide a convincing demonstration of the efficiency of their method (Section 4 and Appendix E).

I am convinced that both contributions would be of interest for TMLR's audience.

**Broader Impact Concerns:**

No comments required in my opinion.

**Claims And Evidence:**

Yes

**Claims Explanation:**

- One of the key strengths of the manuscript is that the authors provide a theoretical proof of the convergence rate of their novel algorithm  and also extensive  numerical experiments related to the optimization of machine learning models. Comments related to numerical experiments are convincing and numerical  results support the theoretical claims related to the main theorems of the manuscript.

- To improve the quality of the manuscript, I have proposed three points to address. The main goal is to explain if the proposed algorithm is of large applicability or not. These are important points to address for increasing the impact of the manuscript.

**Requested Changes:**

Critical
========
- The authors should provide additional comments on the choice of the restriction and prolongation operators based on the literature related to multilevel methods for optimization problems.

- The authors should explain more carefully how they determine in practice the size of the coarse problem $N$ and the order of the low rank approximation of the reduced Hessian matrix $p$.
Both parameters have a strong impact on the convergence rate of SigmaSVD and detailed comments are expected at the theoretical level in the core of the manuscript.

- Additional comments are required concerning the use of randomized methods as discussed above. See the PhD thesis of Nathan Epperly (Caltech, 2025) "Make the Most of What You Have: Resource-Efficient Randomized Algorithms for Matrix Computations" for an accurate overview of the latest contributions in this field.

Less critical
=========
- The authors should incorporate in Section 4 or Appendix E comments related to the computational cost of SigmaSVD in practice. It would be of interest to know the different contributions in terms of CPU or GPU time.

- The bibliography should be extended by considering latest contributions in the field. In the current version of the manuscript, no references on multilevel methods for optimization problems dated later than 2023 are provided.

- To improve readability, Figures 1, 3, 4 and 5 should be larger.

- Section 3.2, page 8, please use "SigmaSVD will converge..."
- Section 3.3, page 8, please use "Polyak-Lojasiewicz" instead of "PL" in the title of this section
- Section 4.1, page 10, please use "selected as 0.5."
- Section 4.1, page 10, please use "In Figures 1a, 1c ..."
- Section 4.2, page 12, please use "one with  N = 2,800..."
- Section 4.2, page 12, please use "another with  N = 1,400..."
- Section 4.2, page 12, please use "as observed in Figure 1c."
- References, please check if arXiv references have been published.
- Appendix B, page 16, please use " where $Q_{h,k}^{-1}$ is defined in equation 10".
- Proof of Lemma B.1, page 17, please replace at several places $\nabla f^2(x_k)$ with $\nabla^2 f(x_k)$.
- Proof of Lemma B.1, page 17, please use $\nabla f(x_k)$ instead of $\nabla f(x_k)^T$ in the last term of the right-hand side in the first equation of the proof.
- Proof of Lemma B.9, page 19, equation just before relation (32), please use $I_{n \times n}$ instead of $I_{N \times N}$.
- Page 20, just before the proof of Theorem 3.1, please remove "Recall the".
- Page 21, Lemma C.1, please use "Newton decrement".
- Page 23, Lemma C.3, please use "...that arises...".
- Page 24, Lemma C.6, please use $I_{n \times n}$ instead of $I_{N \times N}$.
- Page 25, Appendix D, last relation at the bottom of the page, please use $I_{N \times N}$ instead of $I_{n}$.
- Page 26, Appendix E, third item, please modify "Newton with with".
- Page 28, Appendix E, please use "reduced Hessian matrix".

---

> ### Author Response · Authors · 2025-12-24
>
> We thank the reviewer for their insightful comments and the time invested in reviewing our manuscript. Below, we address the requested changes point-by-point.
>
> ## Critical Requested Changes
> ### 1. Choice of restriction and prolongation operators
>
> We have expanded the discussion on the choice of operators in the manuscript. As noted by the reviewer, several methods for selecting restriction and prolongation operators exist in the literature, including Gaussian matrices, 1- or s-hashing, and sampling matrices (see [1]).
>
> While our framework supports these variants, we emphasize that we prefer constructing these operators using Uniform Sampling (as per Definition 1) primarily for computational efficiency. Uniform sampling allows us to avoid expensive matrix multiplications that are often required by denser randomized projections. We have updated the manuscript to explicitly list these alternative choices while justifying our preference for uniform sampling in the context of low-cost iterations.
>
> ### 2. Regarding the choice of $N$.
>
> The paper experimentally investigates the effect of $N$ on the convergence behavior of SigmaSVD. It is indeed true that, without an appropriate choice of $N$, the method may lose its ability to converge rapidly or to escape saddle-point neighborhoods, as illustrated in Figure 1 and Table 1. The experiments also show that $N$ is a problem-dependent parameter. For problems with low-rank Hessian matrices, $N$ can be chosen very small without compromising the fast convergence rate (see, for example, the experiments on the Leukemia and News20 datasets). For this reason, the manuscript states that the method is more suitable for problems with low-rank Hessians—an assumption that is satisfied in most practical problems, especially in non-convex settings. We will add this clarification in the Introduction and Conclusion.
>
> However, establishing a theoretical lower bound on $N$ is quite involved at this stage. The main theoretical framework for obtaining such a bound, together with a quantitative probability that an iterate is successful, is based on the Johnson–Lindenstrauss (JL) lemma [2] and subsequent works. These results, however, are derived in the classical (Euclidean) norm, and extending them to the local norms employed for self-concordant functions is not straightforward. To the best of our knowledge, the only work that provides such bounds is Newton Sketch, which considers subsampling the square-root Hessian matrix. Its theory, however, is based on the results of [3], which cannot be directly applied to our setting. As a result, we leave the task of establishing similar bounds for the local gradient norms as future work. Nevertheless, JL-type bounds can be applied in Theorem 3.3, since the results there are based on the classical norm. We have added this discussion to the manuscript to make Theorem 3.3 more informative. See Remark 3.1.
>
> As with $N$, the parameter $p$ is determined by tuning, and selecting both $N$ and $p$ small yields a ``low-rank'' version of Newton's method. In the absence of theoretical results for the reasons discussed above, when Definition 1 is employed, these parameters should in practice be selected according to the effective rank of the Hessian at the optimum,
> $R = \frac{\operatorname{tr}(\nabla^2 f(\mathbf{x}^*))}{\lambda_{\max}(\nabla^2 f(\mathbf{x}^*))}$.
>
> The quantity $R$ indicates how many directions really matter near the solution, where most methods slow down significantly. If $R$ is known, in practice $N$ and $p$ should be selected slightly larger and slightly smaller than $R$, respectively. Alternatively, they can be selected close to $\max\{n, R \ln n\}$, as suggested in [3]. We have added a remark that addresses these questions.
>
> ## Less Critical Requested Changes
>
> We agree with the reviewer and are happy to make all the suggested changes to improve the readability of the manuscript.
>
> [1] Coralia Cartis, Jaroslav Fowkes, and Zhen Shao. Randomised subspace methods for non-convex optimization, with applications to nonlinear least-squares. arXiv preprint arXiv:2211.09873, 2022.
>
> [2] W. B. Johnson and J. Lindenstrauss, Extensions of Lipschitz mappings into a Hilbert space,
> Contemp. Math., 26 (1984), pp. 189–206.
>
> [3] M. Pilanci and M. J. Wainwright. Randomized sketches of convex programs with sharp
> guarantees. Technical report, UC Berkeley, 2014. Full length version at arXiv:1404.7203;
> Presented in part at ISIT 2014.

---

### Comment · Action_Editor_7npr · 2025-12-18
**Rebuttal**

Dear authors,

The rebuttal period is over but I don't see any of your responses, and the next phase is for the referees to submit their final recommendation in a little under two weeks.  Was the lack of rebuttal intentional?

-Stephen (action editor for this paper)

---

> ### Comment · Action_Editor_7npr · 2026-01-01
> **Rebuttals are in**
>
> Dear reviewers,
>
> The authors have now given a rebuttal. Reviewers, please take a look (and you should also be able to see each other's comments), and then let's discuss (privately) over the next week. I'll ask an editor-in-chief to reset the deadlines for this paper to give us a little more time.
>
> Best,
> Stephen

---

### Decision · Action_Editor_7npr · 2026-02-05

**Recommendation:** Accept as is

**Audience:**

Yes

**Audience Explanation:**

The paper provides what seems to be a novel method, with theory covering certain situations. Again, there was some debate over the appropriateness of the assumptions given the application area, but the numerics somewhat assuage this worry.

**Claims And Evidence:**

Yes

**Claims Explanation:**

The paper provides convergence theory, as well as numerical experiments. While the reviewers discussed the significance of the theory (in particular, is self-concordancy a relevant assumption in the context of machine learning problems?), there were no flaws in the proof found (though the burden on fact-checking every step of the proof is always on the author, not the reviewers). The numerical experiments were mostly deemed sufficient (there was a suggestion to go beyond MNIST, but this wasn't a sticking point).